EMBO
Molecular Medicine

# STIM1-mediated calcium influx controls antifungal immunity and the metabolic function of non-pathogenic Th17 cells

Sascha Kahlfuss[1],[†] [ID], Ulrike Kaufmann[1],[‡], Axel R Concepcion[1], Lucile Noyer[1], Dimitrius Raphael[1], Martin Vaeth[1],[§], Jun Yang[1], Priya Pancholi[1], Mate Maus[1],[¶], James Muller[1], Lina Kozhaya[2], Alireza Khodadadi-Jamayran[1], Zhengxi Sun[1], Patrick Shaw[1],[††], Derya Unutmaz[2], Peter B Stathopulos[3] [ID], Cori Feist[4], Scott B Cameron[5], Stuart E Turvey[5] & Stefan Feske[1],[*] [ID]

## Abstract

Immunity to fungal infections is mediated by cells of the innate and adaptive immune system including Th17 cells. $Ca^{2+}$ influx in immune cells is regulated by stromal interaction molecule 1 (STIM1) and its activation of the $Ca^{2+}$ channel ORAI1. We here identify patients with a novel mutation in STIM1 (p.L374P) that abolished $Ca^{2+}$ influx and resulted in increased susceptibility to fungal and other infections. In mice, deletion of STIM1 in all immune cells enhanced susceptibility to mucosal *C. albicans* infection, whereas T cell-specific deletion of STIM1 impaired immunity to systemic *C. albicans* infection. STIM1 deletion impaired the production of Th17 cytokines essential for antifungal immunity and compromised the expression of genes in several metabolic pathways including Foxo and HIF1α signaling that regulate glycolysis and oxidative phosphorylation (OXPHOS). Our study further revealed distinct roles of STIM1 in regulating transcription and metabolic programs in non-pathogenic Th17 cells compared to pathogenic, proinflammatory Th17 cells, a finding that may potentially be exploited for the treatment of Th17 cell-mediated inflammatory diseases.

**Keywords** $Ca^{2+}$ channel; *Candida albicans*; immunodeficiency; STIM1; Th17 cells

**Subject Categories** Immunology; Microbiology, Virology & Host Pathogen Interaction

## Introduction

Over 150 million people worldwide are estimated to suffer from fungal diseases, with the severity ranging from asymptomatic-mild to life-threatening systemic infections resulting in ~1.6 million deaths associated with fungal disease each year (Bongomin *et al*, 2017). *Aspergillus, Candida, Cryptococcus* species, and *Pneumocystis jirovecii* are the main fungal pathogens responsible for the majority of serious fungal disease cases. *Candida* species are part of the normal human microflora of the gastrointestinal and reproductive tracts in 50–80% of healthy individuals, but can become pathogenic in immune compromised hosts (Brown *et al*, 2012). Common causes of increased susceptibility to *Candida* infections include HIV/AIDS, immunosuppressive therapies, antibiotic use, and inherited immunodeficiencies (Lanternier *et al*, 2013; Bongomin *et al*, 2017; Mengesha & Conti, 2017). Infections with *Candida (C.) albicans* manifest as mucosal or mucocutaneous candidiasis, onychomycosis or systemic fungal infection. Systemic *C. albicans* infection can occur after dissemination of local fungal infections or as nosocomial, often catheter-associated, infections in patients receiving critical care (Villar & Dongari-Bagtzoglou, 2008; Lanternier *et al*, 2013). Immunity to *C. albicans* infections involves innate and adaptive immune responses (Hernandez-Santos & Gaffen, 2012; Conti & Gaffen, 2015; Netea *et al*, 2015; Sparber & LeibundGut-Landmann, 2015). *C. albicans* is initially recognized by cells of the innate immune system including dendritic cells, macrophages, and neutrophils. At skin and mucosal surfaces, *C. albicans* hyphae may enter epithelial cells resulting in their activation and production of

1 Department of Pathology, New York University Grossman School of Medicine, New York, NY, USA
2 The Jackson Laboratory for Genomic Medicine, Farmington, CT, USA
3 Department of Physiology and Pharmacology, Schulich School of Medicine and Dentistry, Western University, London, ON, Canada
4 Department of Obstetrics & Gynecology, Oregon Health & Science University, Portland, OR, USA
5 Division of Allergy and Clinical Immunology, Department of Pediatrics, University of British Columbia, Vancouver, BC, Canada
 *Corresponding author. Tel: +1 212 263 9066; E-mail: stefan.feske@nyulangone.org
 †Present address: Institute of Molecular and Clinical Immunology, Health Campus Immunology, Infectiology and Inflammation, Otto-von-Guericke University Magdeburg, Magdeburg, Germany
 ‡Present address: Genentech, South San Francisco, CA, USA
 §Present address: Institute for Systems Immunology, Julius Maximilians University of Wuerzburg, Wuerzburg, Germany
 ¶Present address: Institute for Research in Biomedicine (IRB), Barcelona, Spain
 ††Present address: Bristol-Myers Squibb, Princeton, NJ, USA

IL-1β, TNF-α, and IL-6, which activate neutrophils and other innate immune cells. The recruitment and activation of neutrophils also depend on TNF-α, IFN-γ, and IL-17A produced by Th1, Th17 cells, type 3 innate lymphoid cells (ILC3) as well as NK cells and γδ T cells (Bar *et al*, 2014; Conti & Gaffen, 2015; Netea *et al*, 2015). Neutrophils are required for clearing fungal pathogens, and *C. albicans* is among the most frequently isolated pathogens in neutropenic patients with nosocomial systemic candidiasis (Delaloye & Calandra, 2014).

On the adaptive side of the immune system, non-pathogenic Th17 cells are critical for antifungal immunity as shown by studies in mice and human patients with inherited defects in Th17 cell differentiation and/or function (Mengesha & Conti, 2017). Individuals with mutations in IL-17A, IL-17 receptor A (IL-17RA), or IL-17RC (Puel *et al*, 2011; Ling *et al*, 2015; Levy *et al*, 2016) are susceptible to chronic mucocutaneous candidiasis (CMC) as are patients with dominant-negative mutations in the transcription factor signal transducer and activator of T cells (STAT) 3 (Milner *et al*, 2008) and gain-of-function (GOF) mutations in STAT1 (Toubiana *et al*, 2016), which result in defects of Th17 cell differentiation. Furthermore, neutralizing autoantibodies to IL-17A, IL-17F, and IL-22 are associated with an increased susceptibility to *C. albicans* infections in patients with autoimmune polyglandular syndrome type 1 (APS1) due to mutations in autoimmune regulator (AIRE) (Kisand *et al*, 2010). In mice, deletion of IL-23p19, which is required for the differentiation of Th17 cells, or the IL-17A receptor (IL-17RA) causes severe oropharyngeal candidiasis (OPC; Conti *et al*, 2009). By contrast, *Il-12⁻/⁻* mice lacking Th1 cells do not develop OPC but fail to prevent *Candida* dissemination to the kidney. Besides immunity to local candidiasis, non-pathogenic Th17 cells are also crucial for immunity to systemic *Candida* infection (Huang *et al*, 2004). Mice lacking IL-17RA had a *C. albicans* dose-dependent survival defect after systemic infection (Huang *et al*, 2004). Intestinal colonization of mice with *C. albicans* was recently shown to mediate Th17 cell differentiation and expansion, which provides immunity to systemic *Candida* infection (Shao *et al*, 2019). Collectively, these studies show that IL-17 signaling is critical for immunity to local *Candida* infection and identify Th17 cells as critical during systemic candidiasis.

The function of T cells depends on calcium ($Ca^{2+}$) influx and signaling (Feske, 2007). The main mode of $Ca^{2+}$ influx in T cells is store-operated $Ca^{2+}$ entry (SOCE) that is mediated by $Ca^{2+}$ release-activated $Ca^{2+}$ (CRAC) channels. CRAC channels are hexamers of ORAI1 proteins located in the plasma membrane that are activated by STIM1 and its homologue STIM2. STIM1 and STIM2 are single-pass membrane proteins located in the endoplasmic reticulum (ER) membrane. T-cell receptor stimulation results in the production of inositol (1,4,5) trisphosphate (IP₃) and opening of IP₃ receptors in the ER membrane, followed by $Ca^{2+}$ release from the ER. This triggers a conformational change of STIM proteins and their binding to ORAI1 in the plasma membrane, resulting in CRAC channel opening and SOCE (Deng *et al*, 2009; Maus *et al*, 2015). Loss-of-function (LOF) mutations in *ORAI1* or *STIM1* genes (OMIM 610277 and 605921) abolish SOCE and cause CRAC channelopathy, which is characterized by combined immunodeficiency (CID), humoral autoimmunity, and ectodermal dysplasia (Lacruz & Feske, 2015; Concepcion *et al*, 2016). CID typically presents in early infancy and can be severe resulting in death of patients from viral and bacterial

infections (Feske *et al*, 2006, 2005; McCarl *et al*, 2009; Lacruz & Feske, 2015). A common feature of CRAC channelopathy are fungal infections with a variety of pathogens including, most commonly, *C. albicans*, *Aspergillus fumigatus,* and *Pneumocystis jirovecii* (Table 1).

We previously reported that the function of pathogenic Th17 cells, which orchestrate inflammation in many autoimmune diseases, depends on CRAC channels. Mice with T cell-specific deletion of ORAI1, STIM1, or both STIM1 and STIM2 had profound defects in Th17 cell function resulting in partial or complete protection from experimental autoimmune encephalomyelitis (EAE), a Th17 cell-dependent murine model of multiple sclerosis (MS; Ma *et al*, 2010; Schuhmann *et al*, 2010; Kaufmann *et al*, 2016). Furthermore, pathogenic Th17 cells that differentiate under the influence of a hyperactive form of STAT3 require STIM1 for their function and ability to cause multiorgan inflammation (Kaufmann *et al*, 2019). By contrast, the role of CRAC channels in non-pathogenic Th17 cells and their ability to mediate immunity to infection with bacterial and fungal pathogens is unknown. In addition, the cause of impaired antifungal immunity in CRAC channel-deficient patients remains elusive.

In this study, we report patients with a novel LOF mutation in STIM1 (p.L374P) that abolishes STIM1 function and SOCE. Like many other SOCE-deficient patients, they suffer from chronic fungal infections. Using mice with conditional deletion of STIM1, we demonstrate that SOCE in non-pathogenic Th17 cells is essential for antifungal immunity to systemic infection with *C. albicans*. Mechanistically, the lack of functional STIM1 and SOCE in non-pathogenic Th17 cells resulted in impaired production of IL-17A and other cytokines, but did not impair the expression of Th17 signature molecules including RORγt. An unbiased transcriptome analysis of non-pathogenic Th17 cells revealed that STIM1 regulates two key metabolic pathways, aerobic glycolysis, and mitochondrial oxidative phosphorylation (OXPHOS), whose function was impaired in STIM1-deficient non-pathogenic Th17 cells. In pathogenic Th17 cells, by contrast, STIM1 only controlled OXPHOS but not glycolysis. The greater reliance of pathogenic Th17 cells on STIM1 and OXPHOS may be exploitable for the treatment of autoimmune diseases in which pathogenic Th17 cells play an important role without affecting the antimicrobial function of non-pathogenic Th17 cells.

# Results

### Mutation of p.L374 in the C terminus of STIM1 abolishes SOCE and causes CRAC channelopathy

We here report two patients (P1, P2) born to parents who are first-cousins-once-removed. The patients presented with combined immune deficiency (CID) and recurrent infections since childhood with varicella zoster virus (VZV), pneumonias caused by atypical mycobacteria, onychomycosis, and skin infections with *C. albicans* species (Fig 1A, Table EV1). Patients also had non-immunological symptoms including congenital muscular hypotonia and anhidrotic ectodermal dysplasia, which are typical of CRAC channelopathy due to LOF mutations in *STIM1* and *ORAI1* (Lacruz & Feske, 2015). Detailed case reports are provided in the Appendix. Laboratory

analyses demonstrated overall normal immune cell populations in both patients. P1 had a profound defect in T-cell proliferation after T-cell receptor (TCR) stimulation with several viruses and *C. albicans* whereas proliferative responses to mitogens (PHA, PWA) were normal (Appendix Table S1). Sequencing of genomic DNA (gDNA) of P1, P2, and their asymptomatic mother revealed a homozygous missense mutation (c.1121T>C) within exon 8 of *STIM1* in both patients, whereas the mother was heterozygous for the same mutation (Fig 1B). gDNA of the father and a sister of P1 and P2, who are both asymptomatic, was not available for analysis. The *STIM1* c.1121T>C mutation is predicted to cause a single amino acid substitution (p.L374P) in the second coiled-coil domain (CC2, aa 345–391) within the C terminus of STIM1. Analysis of 71,702 genomes from unrelated individuals using the gnomAD v3 database (GRCh38) (https://gnomad.broadinstitute.org) demonstrated that the p.L374P mutation of *STIM1* is extremely rare with an allele frequency of 6.98e-6 (1 allele out of 143,330). As CRAC channelopathy due to mutations in *ORAI1* or *STIM1* genes follows an autosomal recessive inheritance, the homozygous *STIM1* c.1121T>C mutation detected in both patients is compatible with their disease. The scaled CADD score of the STIM1 c.1121T>C, p.L374P mutation is 28.5, indicating that it is within the top 0.1% of the most deleterious variants in the human genome.

The c.1121T>C mutation does not interfere with STIM1 mRNA expression as transcript levels were comparable in PBMC from both patients, their mother, and a healthy donor (HD) control either before and after stimulation with anti-CD3/CD28 (Fig 1C). No compensatory upregulation of STIM2 was observed in P1 and P2. Intracellular staining of $CD4^+$ and $CD8^+$ T cells from P1, P2, their mother and a HD also showed comparable STIM1 protein expression (Fig 1D). The specificity of STIM1 staining was verified using fibroblasts of a patient homozygous for a *STIM1* p.E128RfsX9 frameshift mutation that abolishes STIM1 protein expression (Appendix Fig S1A) (Picard *et al*, 2009). Similar STIM1 protein expression in T cells of P1, P2, their mother, and two HD controls was confirmed by Western blot analysis (Fig 1E). Despite normal STIM1 mRNA and protein expression, SOCE in PBMC isolated from P1 and P2 was severely impaired after stimulation with the sarcoplasmic/endoplasmic $Ca^{2+}$ ATPase inhibitor thapsigargin (TG) compared to PBMC from the patients' mother and a HD control (Fig 1F). Abolished SOCE was also observed in T cells from P1 and P2 cultured *in vitro* compared to those of a HD (Appendix Fig S1B). SOCE in the mother's T cells was reduced, likely because she is a heterozygous carrier of the STIM1 p.L374P mutation.

### STIM1 p.L374P mutation abolishes STIM1 puncta formation and co-localization with ORAI1

Because SOCE is strongly impaired but STIM1 protein expression is normal, we hypothesized that the p.L374P mutation affects STIM1 protein function. Leucine 374 is located in the second coiled-coil (CC2) domain of the cytoplasmic C-terminal segment of STIM1. CC2 is part of a functional domain that is necessary and sufficient for

**Table 1. Synopsis of patients with CRAC channelopathy due to loss-of-function mutations in *ORAI1* or *STIM1* and associated fungal infections.**

| Gene | Mutation | Fungal infections | | References |
|------|----------|-------------------|---|------------|
| | | Local | Systemic | |
| *ORAI1* | p.V181SfsX8 | *Pneumocystis jirovecii* pneumonia | n.r. | Lian *et al* (2018) |
| | p.L194P | n.r. | *C. albicans* sepsis | Lian *et al* (2018) |
| | p.G98R | *Aspergillus fumigatus, Candida albicans* | n.r. | Lian *et al* (2018) |
| | p.R91W | Oral candidiasis, gastrointestinal candidiasis | n.r. | Feske *et al* (1996), Feske *et al* (2000), McCarl *et al* (2009) |
| | p.A88SfsX25 | n.r. | n.r. | Partiseti *et al* (1994), McCarl *et al* (2009) |
| | p.A103E/ p.L194P | n.r. | n.r. | Le Deist *et al* (1995), McCarl *et al* (2009) |
| | p.H165PfsX1 | n.r. | n.r. | Chou *et al* (2015) |
| | p.R270X | *Pneumocystis jirovecii* pneumonia | | Badran *et al* (2016) |
| | p.I148S | *Pneumocystis jirovecii* pneumonia, oral thrush, diaper rash | | Klemann *et al* (2017) |
| *STIM1* | p.E128RfsX9 | n.r. | n.r. | Picard *et al* (2009) |
| | C1538-1 G>A | n.r. | n.r. | Sahin *et al* (2010), Byun *et al* (2010) |
| | p.R429C | n.r. | n.r. | Fuchs *et al* (2012), Maus *et al* (2015) |
| | p.R426C | n.r. | n.r. | Wang *et al* (2014) |
| | p.P165Q | n.r. | n.r. | Schaballie *et al* (2015) |
| | p.L74P | n.r. | n.r. | Parry *et al* (2016) |
| | p.L374P | Diaper rash, Onychomycosis[a] | n.r. | This study |

n.r., not reported.
[a]Present prior to, but aggravated by, treatment with infliximab and inhalative corticosteroids.

STIM1 binding to ORAI1 and that is alternatively referred to as CRAC activation domain (CAD, aa 342–448) (Park *et al*, 2009), STIM-ORAI activation region (SOAR, 344–442) (Yuan *et al*, 2009), or CC fragment b9 (CCb9, 339–444) (Kawasaki *et al*, 2009; Fig 2A). We modeled the impact of the p.L374P mutation on the structure of the STIM1 CC2 and CC3 domains using available crystal structures of CAD/SOAR/CCb9 and NMR structures of C-terminal STIM1 fragments (Yang *et al*, 2012; Stathopulos *et al*, 2013). Based on the L374M/V419A/C437T triple mutant crystal structure (Yang *et al*, 2012), the human CAD/SOAR/CCb9 domain forms a V-shaped dimer. Our homology model of the dimer places the M374P mutation (red) at the base of each apex formed by CC2 (light blue ribbon) and CC3 (light green ribbon) (Fig 2B). Backbone atom alignment of residues 344–370 with the template structure (gray ribbon) reveals an ~6.4 Å movement of each mutant apex position, which is predicted to perturb STIM1 interaction with ORAI1. Furthermore, we created a homology model of the p.L374P mutation in the context of the solution NMR structure of a human STIM1-ORAI1 dimer complex (Stathopulos *et al*, 2013). The dimer is formed by two STIM1 peptides (aa 312–387) containing the most carboxyl (C) terminal part of CC1 (light pink ribbon) and the entire CC2 (light blue ribbon) in complex with two ORAI1 peptides (aa 272–292) corresponding to the C-terminal CC domain of ORAI1 (Orai1-$C_{272–292}$, yellow ribbons) (Fig 2C). In this model, the p.L374P mutation (red) is located in a critical area of the STIM-ORAI association pocket (SOAP), which confers a hydrophobic surface for each STIM1-ORAI1-$C_{272–292}$ interaction. The loss of a backbone amide hydrogen (NH) by the p.L374P mutation results in an altered hydrogen bond network near the mutation, which is predicted to interfere with STIM1 helix stability, folding, and thus binding to ORAI1. Note that both the crystal and NMR conformations could similarly experience a helical bend due to the rigid nature of the Pro peptide bond and helical destabilization due to the absence of a backbone NH in Pro.

To test whether the p.L374P mutation interferes with STIM1 function and its ability to bind to ORAI1, we investigated its co-localization with ORAI1. We first analyzed the formation of STIM1 puncta by total internal reflection fluorescence microscopy (TIRFM) using HEK293 cells stably expressing GFP-ORAI1 and co-transfected with either wild-type (WT) mCherry-STIM1 or mutant mCherry-STIM1 p.L374P. In non-stimulated cells with replete ER $Ca^{2+}$ stores, WT STIM1 was localized in the ER and outside the TIRFM evanescence field as expected (Fig 2D). Following TG-induced depletion of ER stores WT STIM1 became visible in the TIRFM layer and redistributed into discrete puncta consistent with its localization in ER-PM junctions as reported earlier (Figs 2D, F–G and EV1A–C) (Liou *et al*, 2005; Wu *et al*, 2014). By contrast, the mutant STIM1 p.L374P was already localized near the PM in the majority of cells prior to TG stimulation (70.1 ± 4.9% of cells) but without forming discrete puncta (Figs 2E–G and EV1B and C). After store depletion, STIM1 p.L374P localization at the PM did not further increase and it failed to form puncta. Importantly, WT STIM1 and ORAI1 strongly colocalized in puncta after TG stimulation (Figs 2D and H, and EV1B and C). By contrast, no increase in co-localization was observed between mutant STIM1 p.L374P and ORAI1 above levels seen in cells before ER store depletion (Figs 2E and H, and EV1B and C).

These findings suggested that the STIM1 p.L374P mutant has a partially active configuration that allows it to translocate to the PM

without activating ORAI1 channels and SOCE. Recruitment of STIM1 to ER-PM junctions is known to depend on the release of a lysine (K)-rich polybasic domain (PB) in its far C terminus that promotes STIM1 binding to membrane phospholipids (Fig 2A; Liou *et al*, 2007; Park *et al*, 2009). To test if the localization of STIM1 p.L374P near the PM is dependent on the PB, we transfected cells expressing GFP-ORAI1 with WT or mutant m-Cherry-STIM1 lacking the PB (ΔK) and analyzed the presence of STIM1 at the PM and its co-localization with ORAI1 before and after stimulation with TG by TIRFM. Deletion of the PB in WT STIM1-ΔK had no effect on its TG-inducible recruitment to the PM and ability to form puncta (Fig EV1B and C) consistent with previous findings in cells overexpressing both STIM1-ΔK and ORAI1 (Park *et al*, 2009). Removal of the PB in STIM1 p.L374P-ΔK had no effect on its localization near the PM before or after TG stimulation. An alternative explanation for the constitutive presence of STIM1 p.L374P near the PM is its binding to ORAI1, and we therefore overexpressed WT or mutant STIM1 in HEK293 cells in which ORAI1 had been deleted by CRISPR/Cas9 gene targeting. However, despite the lack of ORAI1, STIM1 p.L374P was detected at the PM and additional deletion of the PB had no effect on this localization (A.R.C., S.F. unpublished observation), suggesting that ORAI1 is not required for the recruitment of mutant STIM1 to the PM. This finding and the fact that STIM1 p.L374P partially colocalized with ORAI1 (Figs 2E and H, and EV1B and C) prompted us to test whether mutant STIM1 can bind to ORAI1. As expected, we found an increased binding of WT STIM1 to ORAI1 after TG stimulation in co-immunoprecipitation experiments (Fig EV1D and E). By contrast, STIM1 p.L374P and ORAI1 already co-immunoprecipitated in unstimulated cells, suggesting that mutant STIM1 retains its ability to bind to ORAI1. Stimulation with TG disrupted this interaction, potentially due to TG-induced activation of endogenous STIM1 or STIM2, which may outcompete the weaker binding of mutant STIM1 to ORAI1. Together, our results indicate that STIM1 p.L374P has a partially active conformation that promotes its recruitment to ER-PM junctions. Although this recruitment is not dependent on ORAI1, mutant STIM1 is able to bind to ORAI1 but fails to activate its channel function and SOCE.

### STIM1 p.L374P mutation impairs T-cell proliferation and cytokine production

As patients with *STIM1* p.L374P mutation suffered from CID, we analyzed the composition of their immune cells. Most immune cell populations were only marginally altered (Appendix Table S1). The number of naïve CD4$^+$CD45RO$^-$CCR7$^+$ T cells was strongly decreased in P1 and P2 compared to their mother and a HD control, whereas the frequencies of CD4$^+$CD45RO$^+$CCR7$^-$ effector memory T cells were increased (S.K., S.F. unpublished observations), which was most likely a consequence of the patients' chronic and recurrent infections (McCarl *et al*, 2009; Lian *et al*, 2018). To analyze whether the *STIM1* p.L374P mutation impairs lymphocyte function, we first assessed the response of T cells isolated from the patients' PBMC to stimulation by anti-CD3 and anti-CD28 crosslinking. Stimulation of HD and maternal CD4$^+$ T cells resulted in the formation of T-cell blasts, whereas no increase in cell size was observed in T cells of P1 and P2 (Fig 3A), consistent with data previously reported for P1 (Vaeth *et al*, 2017a). This defect was mimicked by

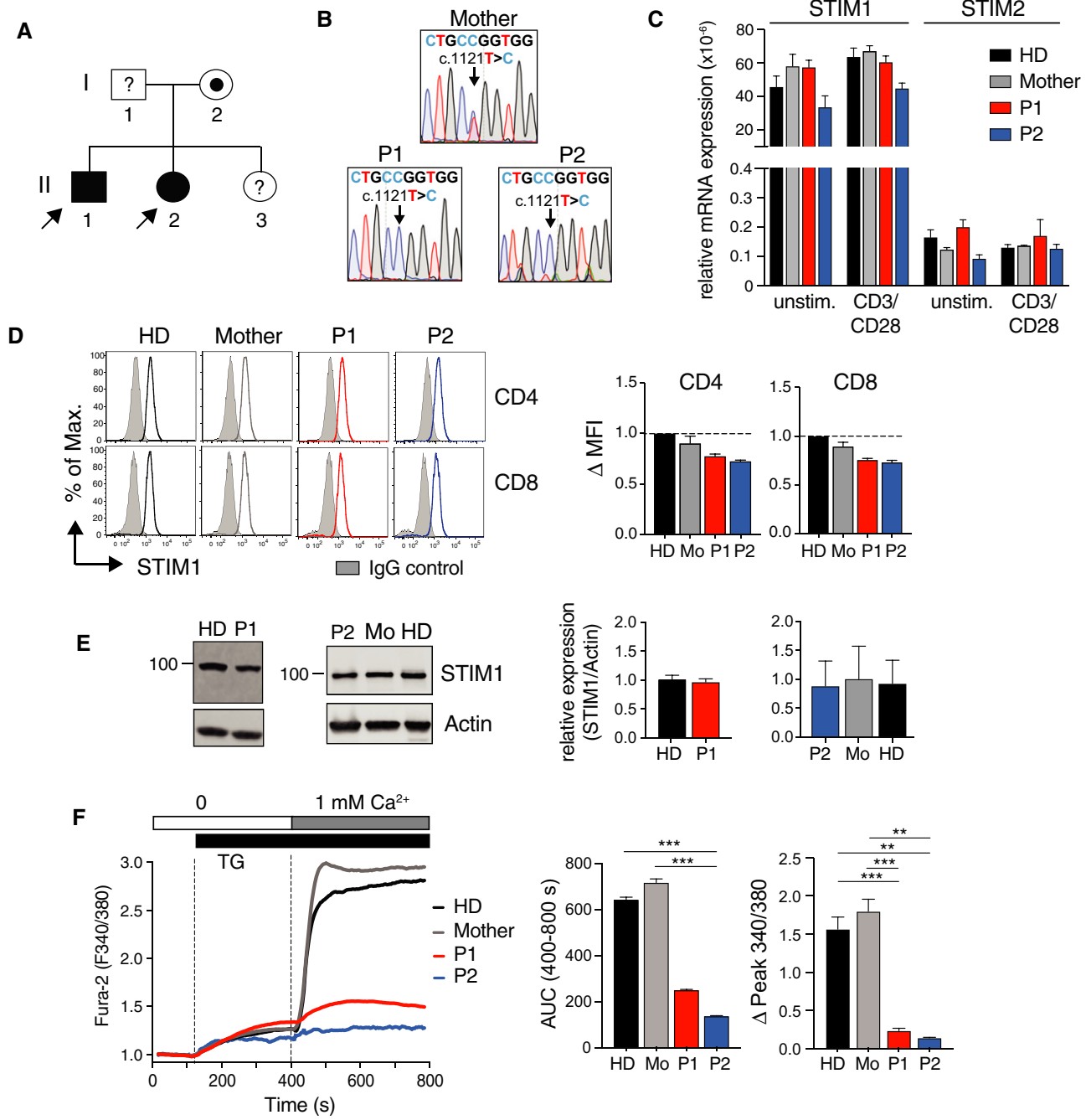

**Figure 1. p.L374P mutation in the C terminus of STIM1 abolishes SOCE and causes CRAC channelopathy.**

A   Pedigree of the patients P1 (II-1) and P2 (II-2) presenting with CRAC channelopathy. Filled circles (females) and squares (males) indicate homozygous patients; dotted symbols indicate heterozygous asymptomatic carriers; symbols with "?" indicate asymptomatic family members not available for DNA sequencing.

B   Sanger sequencing of genomic DNA isolated from PBMC of the mother (I-2), P1, and P2.

C   *STIM1* and *STIM2* mRNA expression in PBMC of P1, P2, mother, and a healthy donor (HD) that were left unstimulated or stimulated with anti-CD3/CD28 for 24 h. mRNA levels of STIM1/2 normalized to 18S rRNA. Graph shows the mean ± SEM of duplicates from one experiment.

D   STIM1 protein expression in CD4$^+$ and CD8$^+$ T cells from P1, P2, the mother, and a HD analyzed by flow cytometry. Shaded histograms: polyclonal rabbit IgG control antibody; open histograms: polyclonal anti-STIM1 antibody. Bar graphs show the delta MFI calculated as MFI$_{STIM1}$ − MFI$_{IgG\ control}$ that was normalized to HD T cells. Bar graphs are mean ± SEM of two independent repeat experiments.

E   STIM1 protein expression in expanded human T cells analyzed by immunoblotting. One representative Western blot of 3 is shown. Bar graphs are the mean ± SEM of STIM1 expression normalized to actin from three independent experiments.

F   Ca$^{2+}$ influx in Fura-2 loaded PBMC of P1, P2, the mother, and a HD. T cells were stimulated with 1 μM thapsigargin (TG) in the absence of extracellular Ca$^{2+}$ followed by addition of 1 mM extracellular Ca$^{2+}$. Bar graphs show the integrated Ca$^{2+}$ influx response (AUC, area under the curve) from 400 to 800 s and the peak Ca$^{2+}$ influx normalized to baseline Ca$^{2+}$ levels (F340/380) at 400 s. Data represent the mean ± SEM from 2 experiments. Statistical analysis by unpaired Student's *t*-test. **$P < 0.01$, ***$P < 0.001$.

Source data are available online for this figure.

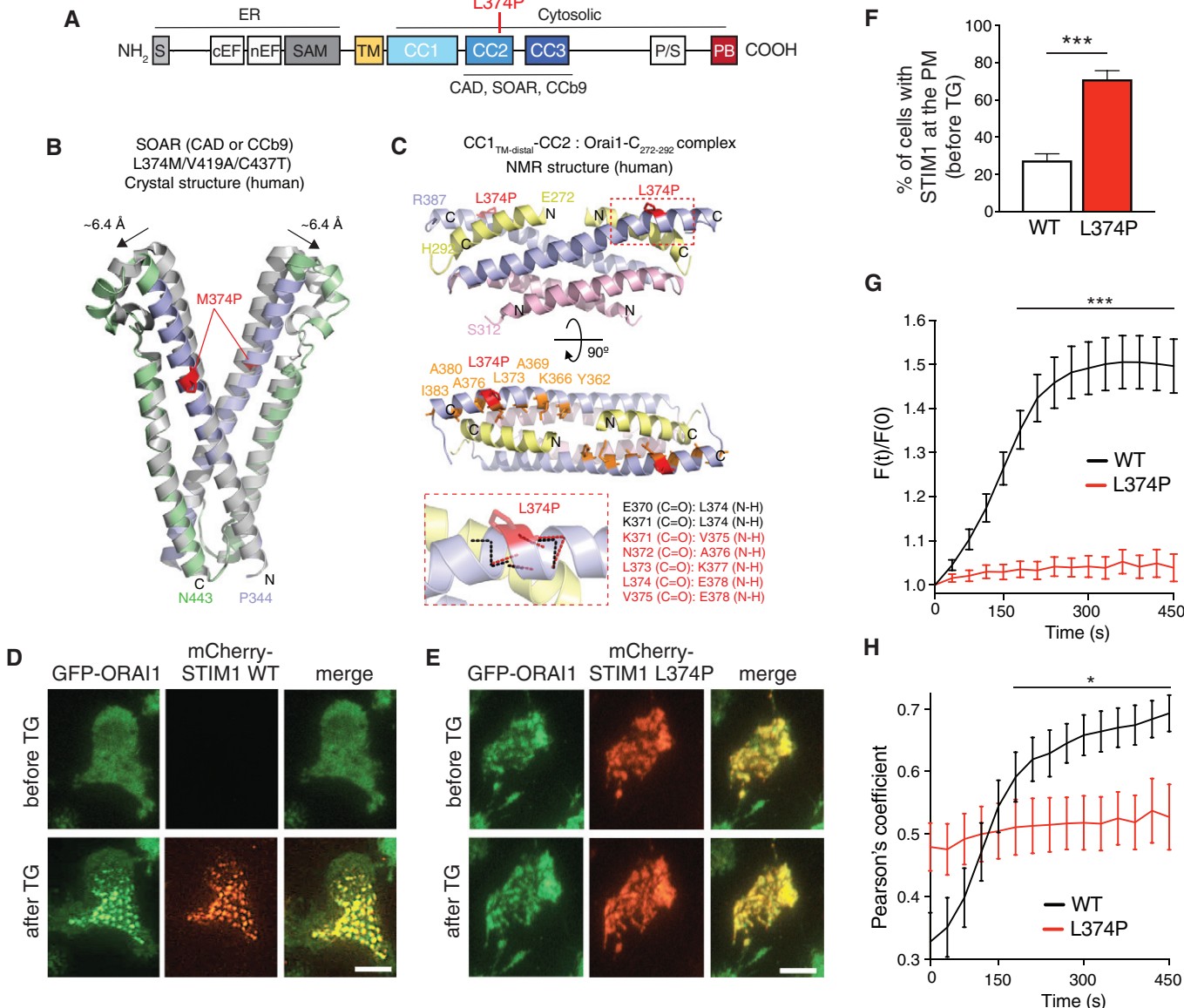

**Figure 2. L374P mutation abolishes STIM1 puncta formation and co-localization with ORAI1.**

A   Domain architecture of STIM1. Abbreviations: CAD: CRAC activation domain; CC1-3: coiled-coil domains 1–3; CCb9, coiled-coil fragment b9; cEF, canonical EF-hand; nEF, non-canonical EF-hand; PB, polybasic domain; P/S, proline/serine-rich region; SAM: sterile alpha motif; S, signal peptide; SOAR: STIM1-ORAI1 activation region; TM: transmembrane domain.

B, C   Homology models of the STIM1 p.L374P mutation based on published crystal (Yang *et al*, 2012) and NMR (Stathopulos *et al*, 2013) structures of STIM1. (B) Location of the M374P mutation (red sticks) in the context of the triple mutant (L374M/V419A/C437T) human CAD/SOAR/CCb9 dimer structure relative to CC2 (light blue ribbon) and CC3 (light green ribbon). The template structure without the M374P mutation (gray ribbon) and the movement of the apex positions of both subunits are indicated. (C) Location of the L374P mutation in the context of the NMR structure of a bimolecular complex of two STIM1 CC1$_{TM\text{-}distal}$-CC2 peptides (CC1$_{TM\text{-}distal}$ region shown as light pink ribbon, and CC2 shown as light blue ribbon) and two ORAI1 C-terminal (aa 272–292) peptides (yellow ribbons) (Stathopulos *et al*, 2013). The loss of the backbone amide hydrogen (NH) due to the L374P mutation results in an altered hydrogen bond network near the mutation (small red dashed box). The zoomed view of this area (large red dashed box) shows the backbone hydrogen bond network for the L374P protein (red dashed lines) compared to the WT protein (black dashed lines). The maintained backbone atom hydrogen bonds in this region for the L374P mutant compared to the WT protein are listed in red text while the lost hydrogen bonds are listed in black text.

D–H   GFP-ORAI1 expressing HEK293 cells were transfected with mCherry-STIM1 (WT or L374P) and left unstimulated or stimulated with 1 μM TG in Ca$^{2+}$-free Ringer's solution. (D,E) Representative TIRF microscopy images from one of two repeat experiments. Scale bars, 10 μm. (F) Percentages of WT or mutant mCherry-STIM1 present at the plasma membrane (i.e., visible in the TIRFM evanescent field) before TG stimulation. Data are the mean ± SEM calculated from 550 (WT) and 830 (L374P) cells from two independent experiments. (G) Change of mCherry-STIM1 (WT or L374P mutant) fluorescence [F(t)/F(0)] in the TIRFM evanescence field. Fluorescence normalized to values at the time of TG addition in Ca$^{2+}$-free Ringer's buffer at 0 s. Traces show mean ± SEM of 40–50 cells from two independent experiments. (H) Co-localization of mCherry-STIM1 (WT or L374P mutant) and GFP-ORAI1 from the time of TG addition at 0 s. Co-localization measured using Pearson's coefficient for GFP and mCherry. Traces show mean ± SEM of 27 cells from two independent experiments.

Data information: Statistical analysis by unpaired Student's *t*-test (F, G) and Mann–Whitney U-test (H). *$P < 0.05$; ***$P < 0.001$.

stimulation of HD and maternal T cells in the presence of FK506 (tacrolimus), which inhibits the $Ca^{2+}$-dependent phosphatase calcineurin. Stimulation of $CD4^+$ and $CD8^+$ T cells by anti-CD3/CD28 crosslinking resulted in robust proliferation of the mother's and HD's cells, whereas the majority of T cells of P1 and P2 did not divide (Fig 3B). A similar or even more pronounced proliferation defect was observed in T cells of P1, P2, mother, and HD after treatment with FK506. It is noteworthy that no proliferation defect is observed when T cells of P1 and P2 are first stimulated with PHA and irradiated allogeneic cells and restimulated with anti-CD3/CD28 (Appendix Fig S2), suggesting that extensive costimulation can bypass the TCR and SOCE dependency of T-cell proliferation. This is consistent with our previous report that SOCE is critical for the metabolic reprogramming of naïve T cells and cell cycle entry after initial TCR stimulation and the ability of IL-2 and IL-7 to bypass this SOCE requirement (Vaeth *et al*, 2017a). We next measured cytokine expression by $CD4^+CD45RO^+$ memory T cells within PBMC isolated from P1, P2, their mother, and a HD after stimulation with PMA and ionomycin *in vitro*. The production of IFN-γ, TNF-α, GM-CSF, and the Th17 cytokines IL-17A and IL-22 was strongly reduced in T cells of P1 and P2 compared to maternal and HD T cells (Fig 3C and D). By contrast, IL-4 production was comparable in patients and controls. Expression levels of IL-2, IFN-γ, and IL-17A cytokines were also impaired in $CD4^+$ and $CD8^+$ T cells of P1 and P2 that had been cultured *in vitro* (Appendix Fig S3A and B). Collectively these findings demonstrate that STIM1 function and SOCE are required for the growth, proliferation, and cytokine production of T cells.

### STIM1-deficient human Th17 cells primed with *Candida albicans* fail to produce IL-17A

Combined immune deficiency in P1 and P2 was associated with fungal infections including severe onychomycosis in P2 and to lesser degree in P1 (Fig 4A, Table EV1). Local and systemic fungal infections were also reported in other patients with CRAC channelopathy (Table 1), but the immunological mechanisms by which SOCE controls antifungal immunity remain poorly understood. To test how loss of SOCE affects the function of human *C. albicans*-specific Th17 cells, we cultured PBMC of a HD for 6 days in the presence of *C. albicans* and restimulated cells in the presence of the CRAC channel inhibitor GSK-7975A or DMSO (Rice *et al*, 2013). $CD4^+$ and $CD8^+$ T cells treated with GSK-7975A showed a significant defect in IL-17A production compared to untreated cells (Fig 4B). We next tested if T cells from patients with the STIM1 p.L374P mutation have a similar defect in cytokine production. T cells of a HD or the patients' mother that were cultured in the presence of *C. albicans* for several weeks produced robust amounts of IL-17A, whereas T cells of P1 and GSK-7975A-treated T cells from 2 HDs failed to do so (Fig 4C). The extent of SOCE inhibition in T cells of P1 and HD T cells treated with the CRAC channel inhibitor was comparable (Fig 4D). T cells of P1 cultured in the presence of *C. albicans* also failed to produce TNF-α and IFN-γ upon restimulation (Appendix Fig S3C and D). It is noteworthy that IL-22 production was not detectable under these culture conditions. Our findings demonstrate that SOCE is required for the function of Th17 cells in response to *C. albicans* infection.

### Immunity to mucosal *Candida albicans* infection requires SOCE mediated by STIM1 and STIM2

Immunity to mucocutaneous *Candida* infection is dependent on cells of the innate and adaptive immune system such as neutrophils, macrophages, NK cells, mast cells, γδ T cells, and αβ T cells (Netea *et al*, 2015). We tested how complete lack of SOCE by genetic deletion of STIM1 and its homologue STIM2 in all immune cells affects immunity to mucosal *C. albicans* infection. *Stim1^{fl/fl}Vav-iCre* and *Stim1/Stim2^{fl/fl}Vav-iCre* mice, which lack SOCE in lymphoid and myeloid cells, as well as WT control mice were infected sublingually with *C. albicans*. STIM-deficient mice showed a significant loss of body weight and dramatic increase in fungal burdens in their tongues 7 days post-infection (p.i.) compared to WT controls (Fig 5A and B). The tongues of STIM1 and STIM1/2-deficient mice were covered with white plaques indicative of high fungal burdens and histologically showed severe inflammation with neutrophil infiltration and large epithelial lesions (Fig 5C and D, left). Periodic acid–Schiff (PAS) staining revealed "nests" of *C. albicans* in the lamina propria of *Stim1^{fl/fl}Vav-iCre* tongues, whereas the tongues of *Stim1/Stim2^{fl/fl}Vav-iCre* mice were completely infiltrated with large numbers of *C. albicans* hyphae, resulting in the destruction of the normal tongue epithelial architecture (Fig 5C and D, right). The cellular analysis of tongue-infiltrating immune cells demonstrated higher frequencies of neutrophils in *Stim1^{fl/fl}Vav-iCre* and *Stim1/Stim2^{fl/fl}Vav-iCre* mice 7 days p.i. compared to WT controls (Fig 5E). By contrast, the numbers of $CD4^+$ T cells producing TNF-α, IL-17A, GM-CSF, and IFN-γ in the draining LNs of *Stim1^{fl/fl}Vav-iCre* and *Stim1/Stim2^{fl/fl}Vav-iCre* mice were reduced compared to controls (Fig 5F and G). Although the numbers of neutrophils were increased in *Stim1^{fl/fl}Vav-iCre* and *Stim1/Stim2^{fl/fl}Vav-iCre* mice 7 days p.i. (Fig 5E), STIM1 and STIM1/STIM2-deficient neutrophils failed to kill ingested *C. albicans* (Fig 5H) and had reduced ROS production upon stimulation *in vitro* (Fig 5I). Together, these data demonstrate that abrogation of SOCE in all immune cells strongly impairs immunity to mucosal *C. albicans* infection.

### STIM1 in T cells is required for immunity to systemic *Candida albicans* infection

Because of the defect in Th17 cytokine production in P1 and P2 and the known role of Th17 cells in antifungal immunity (Mengesha & Conti, 2017), we analyzed mucosal and systemic antifungal immune responses of mice with conditional STIM1 deletion in T cells, which strongly reduces SOCE (Oh-Hora *et al*, 2008; Desvignes *et al*, 2015). Sublingual infection of *Stim1^{fl/fl}Cd4Cre* mice with $2 \times 10^7$ CFU of *C. albicans* did not result in a sustained loss of body weight (Fig 6A) or a significant increase in *C. albicans* burdens in the tongue 7 days p.i. (Fig 6B) compared to littermate controls. Macroscopically the tongues of *Stim1^{fl/fl}Cd4Cre* mice appeared normal without observable signs of *C. albicans* infection (Fig 6C) and histological analysis by H&E and PAS staining 7 days p.i. showed only moderately more neutrophil infiltration in the lamina propria, small epithelial lesions, and no signs of fungal infiltration compared to tongues of WT mice (Fig 6D). The lack of a more pronounced defect in adaptive immunity to mucosal *C. albicans* infection occurs despite severely impaired cytokine production by T cells in the absence of STIM1 (Fig 5F), which is consistent with reports that T cell-deficient

$Rag1^{-/-}$ mice are only moderately more susceptible to oropharyngeal *C. albicans* infection (Gladiator *et al*, 2013; Conti *et al*, 2014).

At least one patient with CRAC channelopathy developed *C. albicans* sepsis (Table 1; Lian *et al*, 2018), suggesting that SOCE may be required for immunity to systemic fungal infection. We therefore

challenged WT and *Stim1$^{fl/fl}$Cd4Cre* mice with a single *i.v.* injection of *C. albicans* (or PBS as control). The survival rate of WT mice at 20 days p.i. with *C. albicans* was ~ 40% whereas all infected *Stim1$^{fl/fl}$Cd4Cre* mice had died by day 18 (Fig 6E). Infected *Stim1$^{fl/fl}$Cd4Cre* mice lost ~ 20% of their body weight at 6 days p.i.

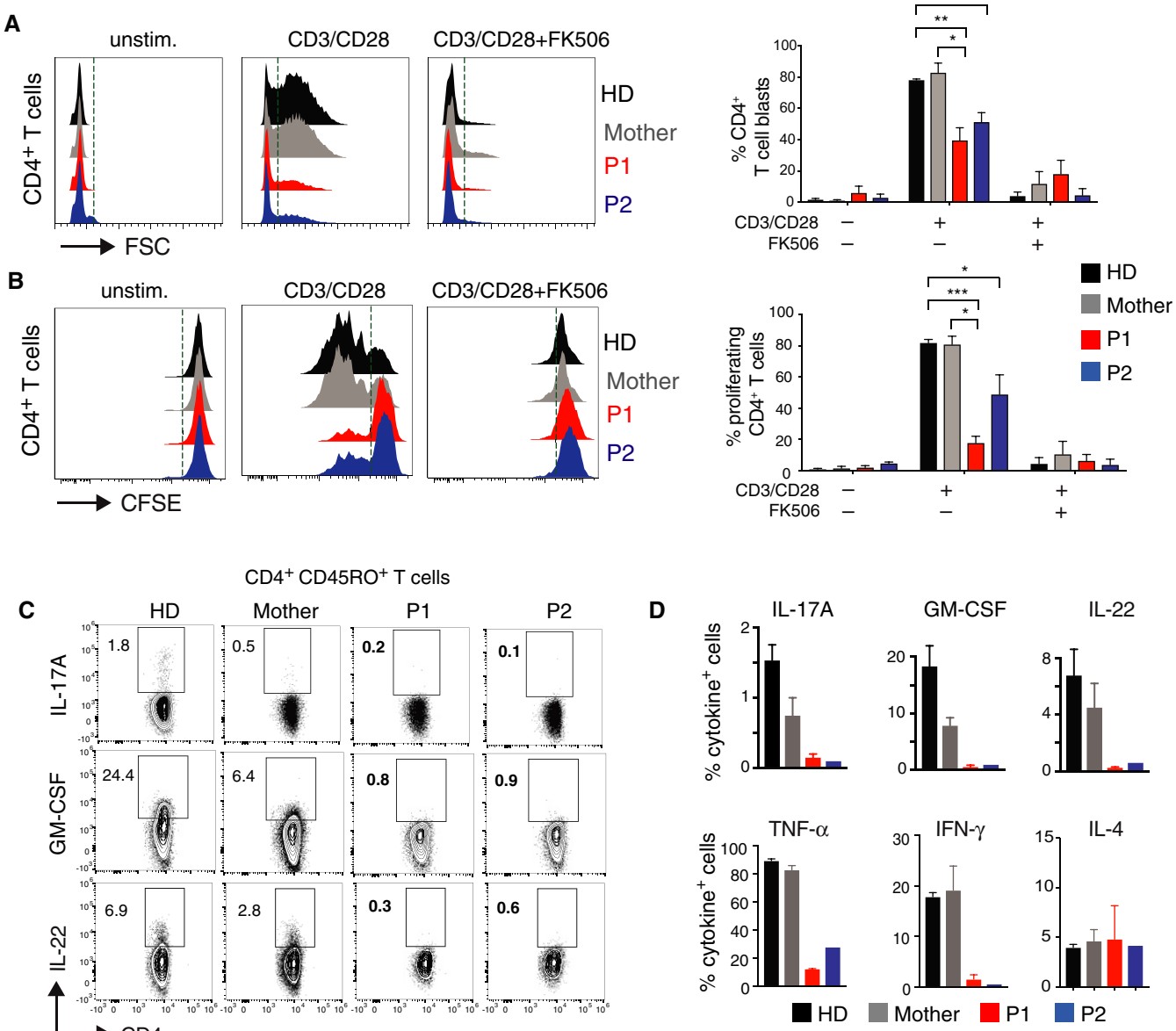

**Figure 3. STIM1 p.L374P mutation causes defect in T-cell proliferation and cytokine production.**

A, B  Cell size (A) and proliferation (B) of CD4⁺ T cells from P1 (red), P2 (blue), their mother (gray), and a HD (black) stimulated with anti-CD3 (5 μg/ml) and anti-CD28 (10 μg/ml) in the presence or absence of 1 μM FK506 for 24 h. (A) Representative histograms of FSC (left panel) and percentages of T-cell blasts (defined as cells to the right of the dotted vertical line) analyzed by flow cytometry (right panel). (B) Representative histograms of CFSE dilution (left panel) and percentages of proliferating cells (defined as cells to the left of the dotted vertical line) (right panel). Bar graphs in A and B are the mean ± SEM from two independent experiments.

C, D  Cytokine production by PBMC from P1, P2, the mother, and an unrelated HD after stimulation with PMA (40 ng/ml) and ionomycin (500 ng/ml) for 4 h. Cytokines were analyzed by flow cytometry following surface staining with antibodies against CD3, CD4, and CD45RO, permeabilization and intracellular cytokine staining for GM-CSF, IL-22, and IL-17A. Representative flow cytometry plots (C) and quantification of Th17 (GM-CSF, IL-22, IL-17A), Th1 (TNF-α, IFN-γ), and Th2 (IL-4) cytokines (D). Data represent the mean ± SEM from two independent experiments.

Data information: Statistical analysis by unpaired Student's *t*-test. *$P < 0.05$, **$P < 0.01$, ***$P < 0.001$.

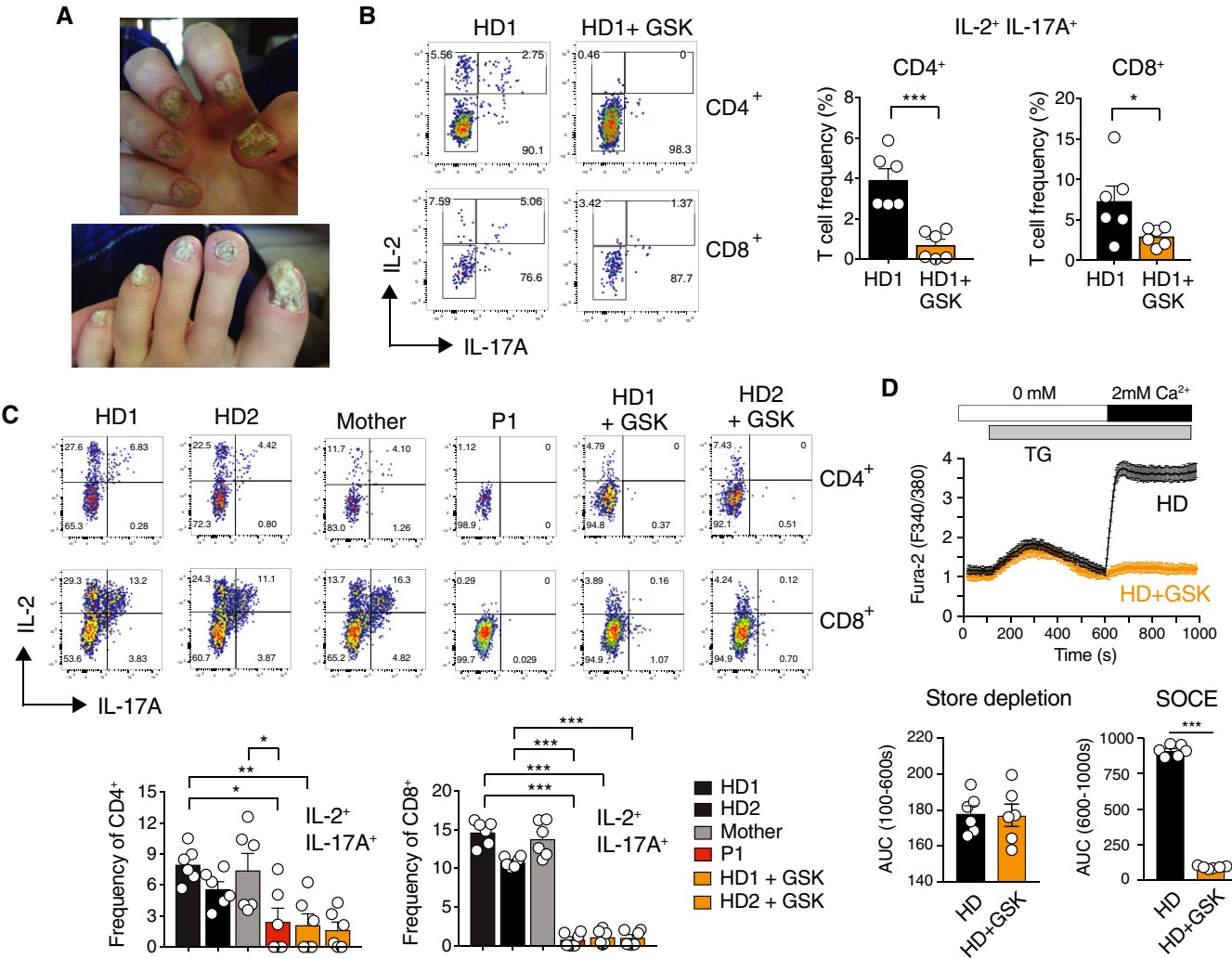

**Figure 4. STIM1 and SOCE mediate IL-17A production of T cells primed with *C. albicans*.**

A    Severe onychomycosis in P2 (*STIM1* p.L374P).

B    PBMC of a healthy donor (HD1) cultured for 6 days in the presence of 6.5 mg/ml *C. albicans* protein. At day 6, PBMC were restimulated with PMA and ionomycin in the presence of 10 μM GSK-7975A or DMSO for 6 h and CD4⁺ and CD8⁺ T cells were analyzed for cytokine production by flow cytometry. Bar graphs represent the mean ± SEM from two independent experiments.

C, D  Human T cells from two healthy donors, P1, and his mother were expanded and cultured in the presence of *C. albicans* and IL-2 for 2–4 weeks. (C) T cells were restimulated with PMA and ionomycin for 6 h in the presence or absence of 10 μM GSK-7975A and analyzed for cytokine production by flow cytometry. Bar graphs represent the mean ± SEM from two independent experiments. (D) Fura-2 loaded T cells were stimulated with thapsigargin (TG) in Ca²⁺-free buffer followed by readdition of 2 mM Ca²⁺ to induce SOCE. Bar graphs represent the mean ± SEM from two independent experiments.

Data information: Statistical analysis in (B) and (D) by unpaired Student's *t*-test and in C by one-way ANOVA. *$P < 0.05$, **$P < 0.01$, ***$P < 0.001$.

compared to ~ 10% in WT littermates (Fig 6F). To assess fungal burdens of different organs during systemic candidiasis, we harvested kidney, liver, and lung of mice 6 days p.i. The kidneys of *Stim1^{fl/fl}Cd4Cre* mice appeared paler than those of WT controls consistent with *C. albicans* dissemination (Fig 6G). The fungal burdens detected in tissue homogenates of kidneys, livers, and lungs from *Stim1^{fl/fl}Cd4Cre* mice were markedly higher at 6 days p.i. compared to WT mice (Fig 6H). H&E staining of kidneys from *C. albicans*-infected *Stim1^{fl/fl}Cd4Cre* mice showed a strong focal infiltration of immune cells that was markedly more pronounced than in WT mice, and PAS staining revealed higher fungal burdens

in *Stim1^{fl/fl}Cd4Cre* kidneys (Fig 6I). To analyze T-cell function after systemic *C. albicans* infection, we isolated splenocytes from WT and *Stim1^{fl/fl}Cd4Cre* mice at 6 days p.i. and restimulated them *ex vivo*. The frequencies of STIM1-deficient CD4⁺ T cells producing IL-2, TNF-α, IFN-γ, and IL-17A were significantly reduced compared to WT controls, whereas the percentage of GM-CSF-secreting CD4⁺ T cells was increased (Fig 6J). GM-CSF functions as a chemoattractant for neutrophils *in vivo* (Khajah *et al*, 2011). Consistent with the enhanced frequencies of GM-CSF⁺ T cells in *Stim1^{fl/fl}Cd4Cre* mice, the frequencies of neutrophils in the blood of *C. albicans*-infected *Stim1^{fl/fl}Cd4Cre* mice were markedly increased 7 days p.i. compared

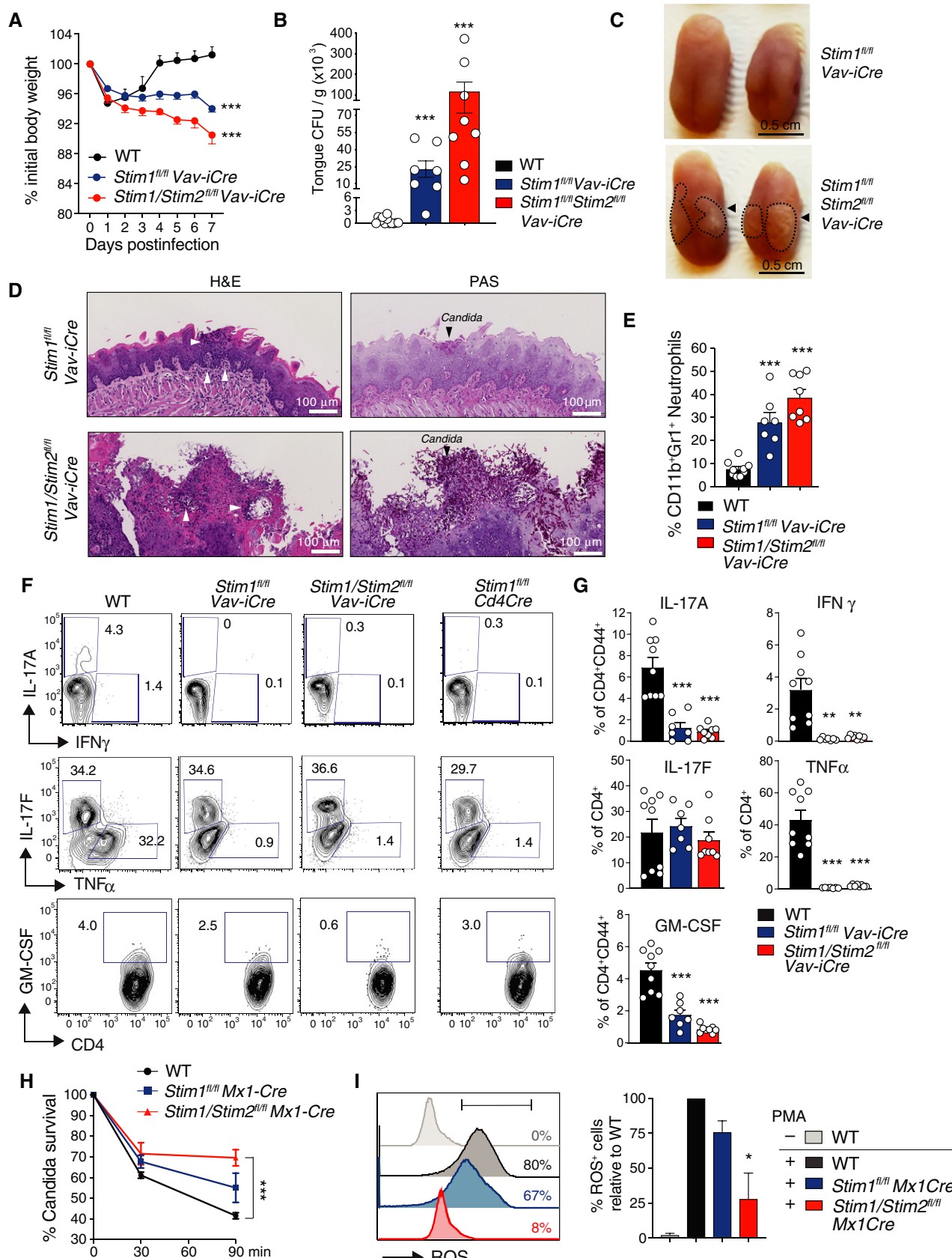

**Figure 5.**

**Figure 5.   STIM1 and STIM2 in immune cells are essential for immunity to mucosal *C. albicans* infection.**

A–G  Sublingual infection of WT, *Stim1^{fl/fl}Vau-iCre*, and *Stim1^{fl/fl}Stim2^{fl/fl}Vau-iCre* mice with $2 \times 10^7$ CFU of *C. albicans*. (A) Body weight (BW) measured for 7 days and normalized to BW at day 0 of infection. Data are mean $\pm$ SEM measured from 4 repeat experiments and 13 WT, 7 *Stim1^{fl/fl}Vau-iCre*, 8 *Stim1^{fl/fl}Stim2^{fl/fl}Vau-iCre* mice. (B) Quantification of CFU of *C. albicans* per gram tongue tissue of the indicated mice at day 7 p.i. Data are mean $\pm$ SEM from 4 repeat experiments and 12 WT, 7 *Stim1^{fl/fl}Vau-iCre*, 8 *Stim1^{fl/fl}Stim2^{fl/fl}Vau-iCre* mice. (C) Macroscopic images of the *C. albicans*-infected tongues of mice at day 7 p.i. Arrows and dashed lines indicate *C. albicans* infiltrated areas. (D) Histological images of H&E (left)- and PAS (right)-stained tongue tissues at day 7 p.i. Clusters of polymorphonuclear (PMN) cells (neutrophils, white arrows) and *Candida* (black arrows) are indicated. (E) Frequencies of Cd11b^{+}Gr1^{+} neutrophils in the tongues from the indicated mice at day 7 p.i. Bar graphs represent the mean $\pm$ SEM from three experiments and 9 WT, 7 *Stim1^{fl/fl}Vau-iCre*, 8 *Stim1^{fl/fl}Stim2^{fl/fl}Vau-iCre* mice. (F,G) Cytokine production by CD4^{+} T cells isolated from submandibular lymph nodes of WT, *Stim1^{fl/fl}Vau-iCre*, *Stim1^{fl/fl}Stim2^{fl/fl}Vau-iCre*, and *Stim1^{fl/fl}Cd4Cre* mice 7 days after sublingual infection with $2 \times 10^7$ CFU *C. albicans*. Representative contour plots of cytokine production (F) and percentages of CD4^{+} T cells producing TNF-$\alpha$, IFN-$\gamma$, IL-17A, IL-17F, and GM-CSF (G). Data are the mean $\pm$ SEM from two independent experiments and 9 WT, 7 *Stim1^{fl/fl}Vau-iCre*, 8 *Stim1^{fl/fl}Stim2^{fl/fl}Vau-iCre* mice.

H, I  Candicidal function and ROS production of neutrophils depend on SOCE. (H) Coculture of Cd11b^{+}Gr-1^{+} neutrophils from poly(I:C) treated WT, *Stim1^{fl/fl}Mx1Cre*, and *Stim1^{fl/fl}Stim2^{fl/fl}Mx1Cre* mice with *C. albicans in vitro* for 0, 30, 60, and 90 min (MOI 0.5). Percentages of live *C. albicans* isolated from neutrophils that were lysed at the indicated time points. Data are the mean $\pm$ SEM from 6 mice per genotype and three independent repeat experiments. (I) ROS production by Cd11b^{+}Gr-1^{+} neutrophils from poly(I:C) treated WT, *Stim1^{fl/fl}Mx1Cre*, and *Stim1^{fl/fl}Stim2^{fl/fl}Mx1Cre* mice was measured after loading of cells with dihydrorhodamine 123 and stimulation with 10 nM PMA for 30 min. Representative histogram plots and bar graphs indicating the percentages of ROS^{+} neutrophils normalized to WT neutrophils stimulated with PMA. Data are the mean $\pm$ SEM from two mice per genotype and two independent repeat experiments.

Data information: Statistical analysis by unpaired Student's *t*-test. *$P < 0.05$, **$P < 0.01$, ***$P < 0.001$.

to WT controls (Fig 6K), whereas neutrophil frequencies in the bone marrow of infected WT and *Stim1^{fl/fl}Cd4Cre* mice were similar (Appendix Fig S4). Together, these data demonstrate that STIM1 in T cells is critical for immunity to systemic fungal infection.

## STIM1 differentially regulates gene expression in pathogenic and non-pathogenic Th17 cells

Non-pathogenic Th17 cells play an important role in antifungal immunity (Netea *et al*, 2015), and our data show that STIM1 controls the production of cytokines promoting antifungal immunity including IL-17A, TNF-$\alpha$, and IFN-$\gamma$. Importantly, non-pathogenic Th17 cells differ in their gene expression (and thus likely their function) from pathogenic Th17 cells that promote inflammation in a variety of autoimmune diseases (Lee *et al*, 2012; Gaublomme *et al*, 2015). We therefore investigated how STIM1 and SOCE affect gene expression in non-pathogenic and pathogenic Th17 cells. We isolated CD4^{+} T cells from *Stim1^{fl/fl}Cd4Cre* mice and WT controls and differentiated them *in vitro* into non-pathogenic Th17 cells (with IL-6, TGF-$\beta$) or pathogenic Th17 cells (IL-23, IL-6, IL-1$\beta$) for 2 days. IL-17A production was strongly impaired in pathogenic and non-pathogenic Th17 cells from *Stim1^{fl/fl}Cd4Cre* mice (Fig EV2A). Transcriptome analysis by RNA sequencing and a subsequent principal component analysis (PCA) showed that non-pathogenic and pathogenic Th17 cells of WT origin were markedly distinct from one another in PC1, with additional variances between WT and *Stim1^{fl/fl}Cd4Cre* Th17 cells found in PC2 (Fig EV2B). The total number of differentially expressed genes (DEG) in WT and STIM1-deficient non-pathogenic Th17 cells was 2,880, whereas in pathogenic Th17 cells this number was 622 (Fig 7A). 387 DEG were shared between non-pathogenic and pathogenic Th17 cells in the absence of STIM1, which included genes like *Foxp3* and *Maf* that were upregulated in non-pathogenic and pathogenic *Stim1^{fl/fl}Cd4Cre* Th17 cells.

We next compared DEG in non-pathogenic Th17 cells to published gene expression signatures of non-pathogenic and pathogenic Th17 cells (Fig 7B; Lee *et al*, 2012; Gaublomme *et al*, 2015). Surprisingly, most non-pathogenic Th17 cell signature genes were upregulated in non-pathogenic *Stim1^{fl/fl}Cd4Cre* Th17 cells, whereas pathogenic Th17 cell signature genes were equally up- and down-regulated (Fig 7B), resulting in an overall bias toward a non-

pathogenic Th17 gene expression signature in the absence of STIM1. Some genes, however, that are essential for Th17 cell differentiation, such as the cytokine receptors *Il1r1* and *Il23r*, and effector functions, including the cytokines *Il17a*, *Il17f*, and *Il21*, were significantly downregulated in the absence of STIM1 (Fig 7C). An exception was *Csf2* whose upregulation in STIM1-deficient non-pathogenic Th17 cells is consistent with the increased GM-CSF expression in CD4^{+} T cells isolated from *Stim1^{fl/fl}Cd4Cre* mice after systemic *C. albicans* infection (Fig 6J). Together, these findings demonstrate that deletion of STIM1 results in a specific reduction of Th17-associated cytokines and cytokine receptors, possibly explaining the impaired antifungal immunity of *Stim1^{fl/fl}Cd4Cre* mice, whereas most non-pathogenic Th17 signature genes (including *Rorc* encoding ROR$\gamma$t) were not impaired or upregulated indicating that the overall identity of non-pathogenic Th17 cells is intact in the absence of STIM1. We also analyzed how deletion of STIM1 affects gene expression in pathogenic Th17 cells. We observed a downregulation of many published pathogenic Th17 signature genes in the absence of STIM1 (Lee *et al*, 2012; Gaublomme *et al*, 2015), whereas non-pathogenic Th17 signature genes were upregulated compared to WT (Fig EV2C). The expression of Th17 cytokines and cytokine receptors was reduced in STIM1-deficient pathogenic Th17 cells, albeit to a lesser degree than in non-pathogenic Th17 cells (Fig EV3A). It is noteworthy that several non-Th17 cytokines were expressed at higher levels in both pathogenic and non-pathogenic Th17 cells in the absence of STIM1 including *Il4*, *Il13*, and *Il9* (Fig EV3B). These data demonstrate that deletion of STIM1 in pathogenic Th17 cells results in a shift toward a non-pathogenic Th17 gene signature and partial loss of pathogenic Th17 cell identity.

## STIM1 controls several metabolic pathways in non-pathogenic Th17 cells

Because STIM1 deletion only impaired the production of Th17 cytokines but did not interfere with the transcriptional identity of non-pathogenic Th17 cells, we further analyzed STIM1 regulated pathways in non-pathogenic Th17 cells that could explain the impaired antifungal immunity of *Stim1^{fl/fl}Cd4Cre* mice. KEGG pathway analyses revealed 34 pathways that were dysregulated in both STIM1-deficient pathogenic and non-pathogenic Th17 cells, which

included Th17 cell differentiation, TCR signaling and JAK-STAT signaling pathways (Fig 7D). In addition, we identified 69 pathways that were uniquely dysregulated in STIM1-deficient non-pathogenic Th17 cells. These included the HIF-1, Foxo1 and phosphatidylinositol signaling pathways, carbon metabolism, and biosynthesis of amino acids (Fig 7D and E). A KEGG network analysis showed that two of these pathways, HIF-1 and Foxo1 signaling, were unique in

the sense that they did not share genes with any of the other dysregulated pathways in non-pathogenic Th17 cells (Fig 7E). Within the Foxo signaling pathway most genes, including *Foxo1* itself, were upregulated in STIM1-deficient non-pathogenic Th17 cells, whereas genes in the HIF-1 signaling pathway were equally up- and downregulated (Fig 7F). HIF1-α positively regulates expression of genes encoding key glycolytic enzymes, whereas Foxo1 negatively

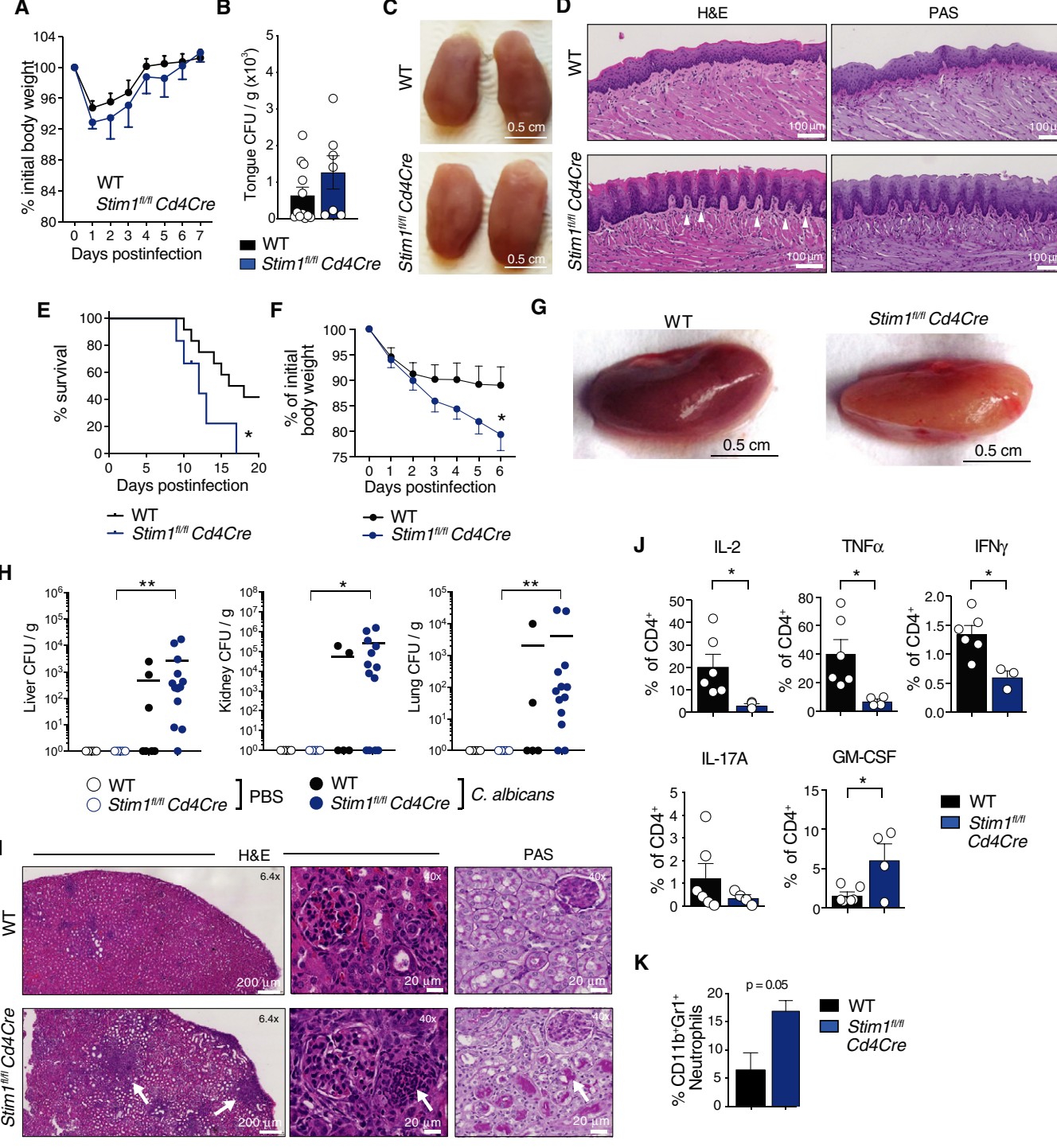

**Figure 6.**

**Figure 6. STIM1 is required for T cell-mediated immunity to systemic fungal infection.**

A–D Sublingual *C. albicans* infection of WT and *Stim1*$^{fl/fl}$*Cd4Cre* mice as described in Fig 5. (A) Body weight (BW) change measured for 7 days p.i. relative to BW at the day of infection. Data are mean ± SEM from 4 repeat experiments and 13 WT and 7 *Stim1*$^{fl/fl}$*Cd4Cre* mice. (B) Quantification of CFU of *C. albicans* isolated from the tongues of mice at day 7 post-infection (p.i.). Data are mean ± SEM from 4 repeat experiments and 7–13 mice per genotype. (C) Macroscopic images of the *C. albicans*-infected tongues of mice at day 7 p.i. (D) Histological images of H&E (left) and PAS-stained (right) tongue tissues at day 7 p.i. Scale bars 100 μm. Arrows point to clusters of polymorphonuclear (PMN) cell (neutrophil) infiltrates.

E–K Systemic *C. albicans* infection. WT and *Stim1*$^{fl/fl}$*Cd4Cre* mice were injected i.v. with $2 \times 10^5$ CFU *C. albicans* or PBS as control. (E) Kaplan–Meier plot showing survival of mice. Data from 5 *Stim1*$^{fl/fl}$*Cd4Cre* mice and 12 WT mice is shown. (F) Body weight (BW) of mice normalized to BW at the day of infection. Data are the mean ± SEM from 4 independent experiments and 8 WT and 7 *Stim1*$^{fl/fl}$*Cd4Cre* mice. (G) Representative macroscopic images of kidneys at day 6 p.i. (H) CFUs of *C. albicans* per gram tissue isolated from the liver, the right kidney, and the lung of mice at day 6 p.i. CFU values where transform to CFU+1 to plot titers on $\log_{10}$ scale. Each dot represents one mouse. Horizontal lines represent mean CFU values. (I) Representative histological images of H&E- and PAS-stained kidneys of mice at day 6 p.i. Magnification 6.4× (left) and 40× (right). White arrows define areas infiltrated with leukocytes (H&E) and *C. albicans* (PAS). (J) Frequencies of CD4$^+$ T cells isolated from the spleens of mice at day 6 p.i. producing the indicated cytokines after restimulation with 20 nM PMA and 1 μM ionomycin for 6 h and analyzed by flow cytometry. (K) Frequencies of CD45$^+$CD11b$^+$Gr-1$^+$ neutrophils in the blood of mice at day 6 post-infection analyzed by flow cytometry. Data in J are the mean ± SEM from two independent experiments and 6 WT and 4 *Stim1*$^{fl/fl}$*Cd4Cre* mice; data in K are the mean ± SEM from one experiment and 2 WT and 3 *Stim1*$^{fl/fl}$*Cd4Cre* mice.

Data information: Statistical analyses in (A), (B), (F), (J) and (K) by unpaired Student's *t*-test; in (E) by log-rank (Mantel–Cox) test and in (H) by Mann–Whitney U-test. Significance levels: *$P < 0.05$, **$P < 0.01$.

regulates the metabolic fitness of CD4$^+$ T cells through downregulation of myc (Buck *et al*, 2017; Newton *et al*, 2018). Gene set enrichment analysis (GSEA) further demonstrated the depletion of myc target genes and genes regulating glycolysis in STIM1-deficient non-pathogenic Th17 cells (Fig 7G). Other metabolic pathways were also downregulated in the absence of STIM1, including genes associated with mitochondrial function and the TCA cycle (Fig 7G). Together these findings indicate that STIM1 controls several metabolic pathways in non-pathogenic Th17 cells including glycolysis through the Foxo- and HIF1α-dependent regulation of myc, as well as TCA cycle function and mitochondrial respiration.

## STIM1 regulates aerobic glycolysis and oxidative phosphorylation in non-pathogenic Th17 cells

Given the dysregulation of several metabolic pathways in non-pathogenic Th17 cells lacking STIM1 and our previous reports that SOCE controls glycolytic gene expression in T cells (Vaeth *et al*, 2017a) and mitochondrial function in pathogenic Th17 cells (Kaufmann *et al*, 2019), we investigated the role of STIM1 in the metabolism of non-pathogenic Th17 cells in more detail. We found significantly reduced mRNA levels of genes encoding glycolytic enzymes in non-pathogenic Th17 cells of *Stim1*$^{fl/fl}$*Cd4Cre* compared to WT mice, which included hexokinase 2 (*Hk2*), phosphofructokinase (*Pfkp*), aldolase A (*Aldoa*), phosphoglycerate kinase 1 (*Pgk1*), enolase (*Eno1*), pyruvate kinase (*Pkm*), and lactate dehydrogenase A (*ldha*) (Fig 8A). Apart from glycolysis, numerous genes encoding enzymes of the TCA cycle (including the rate-limiting isocitrate dehydrogenase homologues *Idh1* and *Idh3a*) and factors promoting mitochondrial respiration such as components of the electron transport chain (ETC) were markedly reduced in STIM1-deficient non-pathogenic Th17 cells compared to WT cells (Fig 8A). We next evaluated the role of STIM1 in the glycolytic function of non-pathogenic Th17 cells. The uptake of the fluorescent glucose analog 2-NBDG (2-(N-(7-nitrobenz-2-oxa-1,3-diazol-4-yl)amino)-2-deoxyglucose) was significantly reduced in STIM1-deficient non-pathogenic Th17 cells (Fig 8B). To measure the glycolytic activity, we analyzed the extracellular acidification rate (ECAR) in the supernatant of WT and STIM1-deficient non-pathogenic Th17 cells. The basal glycolytic rate, glycolytic capacity, and glycolytic reserve were significantly reduced in *Stim1*$^{fl/fl}$*Cd4Cre* non-pathogenic Th17 cells (Fig 8C),

consistent with impaired expression of glycolytic enzymes and glucose uptake (Fig 8A and B). It is important to note that deletion of STIM1 in pathogenic Th17 cells had no effect on the mRNA expression of glucose transporters and glycolytic enzymes (Fig EV3C), which is consistent with our previous findings in a different model of pathogenic Th17 cells (Kaufmann *et al*, 2019) and indicates a crucial difference between the role of SOCE in regulating metabolism in pathogenic and non-pathogenic Th17 cells. Since our GSEA also showed strongly dysregulated expression of genes involved in the TCA cycle and mitochondrial function, we analyzed mitochondrial respiration and OXPHOS by measuring the oxygen consumption rate (OCR) of non-pathogenic Th17 cells. The basal respiration of STIM1-deficient non-pathogenic Th17 cells as well as their maximal respiration and spare respiratory capacity (SRC) measured after uncoupling of mitochondria with FCCP was significantly reduced (Fig 8D). Collectively, our findings demonstrate that STIM1 controls the function of several metabolic pathways in non-pathogenic Th17 cells including aerobic glycolysis and oxidative phosphorylation, which is in contrast to pathogenic Th17 cells in which STIM1 controls OXPHOS but not glycolysis.

## STIM1 regulates glycolysis, OXPHOS, and mTORC1 signaling in human T cells

Given the important role of STIM1 for T-cell metabolism and function in mice, we hypothesized that STIM1 may also regulate the metabolism of human T cells and could thereby be required for their ability to mediate antifungal immunity. To test this hypothesis, we analyzed the glycolytic function and OXPHOS of CD4$^+$ T cells isolated from PBMC of P1 and a HD control that were stimulated with anti-CD3/CD28 in the presence or absence of the calcineurin inhibitor FK506 for 24 h. STIM1-deficient human CD4$^+$ T cells had significantly reduced glycolysis and an all but abolished glycolytic reserve after ATP synthase inhibition with oligomycin (Fig 8E and F). Similar defects were observed in HD CD4$^+$ T cells treated with FK506, suggesting that the STIM1 effects on glycolysis are dependent on calcineurin function. Besides impaired glycolysis, freshly isolated CD4$^+$ T cells of the STIM1-deficient P1 had a severe defect in mitochondrial function. Their basal respiration as well as their spare respiratory capacity, ATP production, and coupling efficiency were strongly reduced compared to CD4$^+$ T cells from a HD (Fig 8G

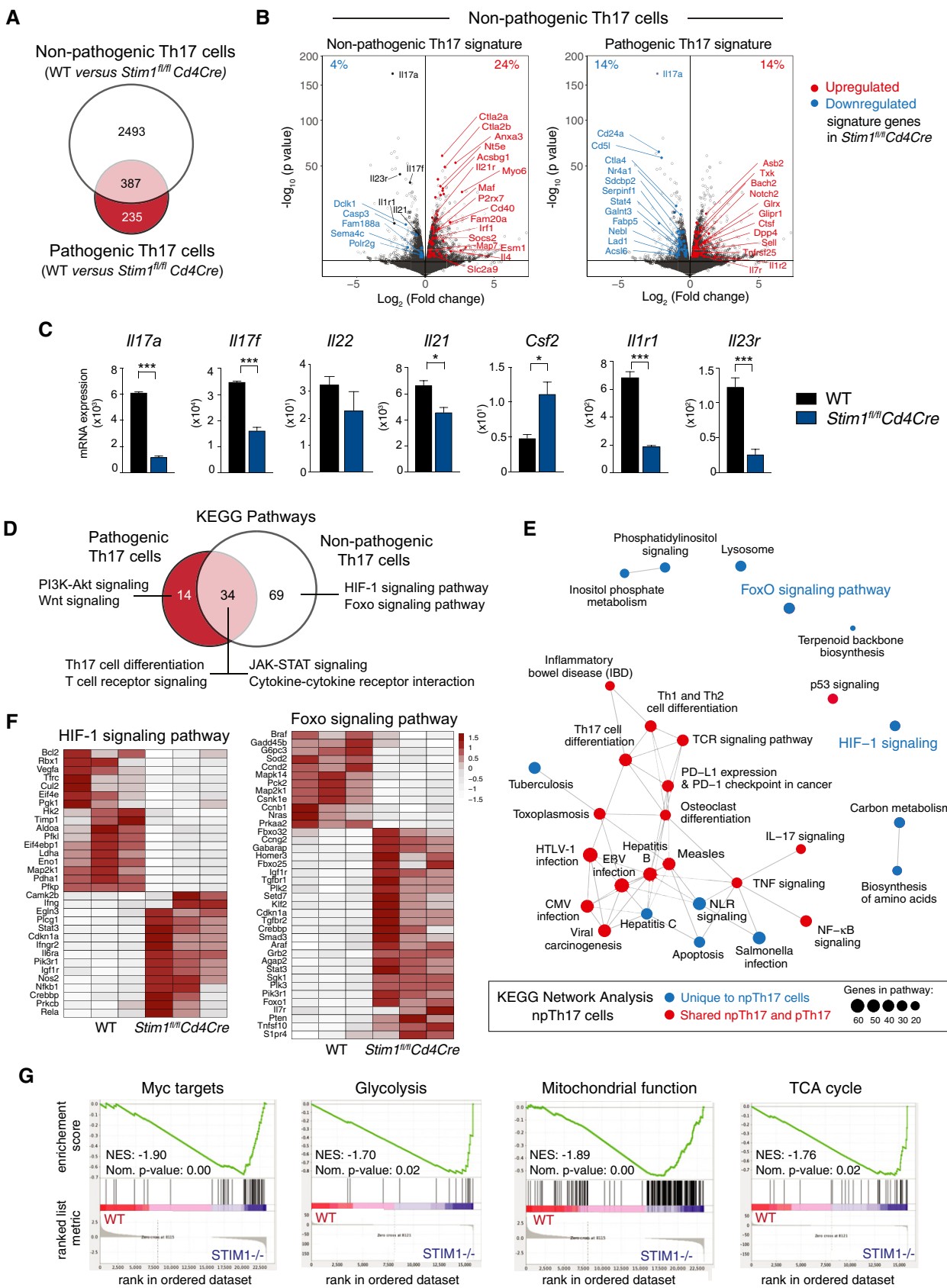

Figure 7.

**Figure 7. STIM1 differentially regulates expression of genes associated with pathogenic and non-pathogenic Th17 cells.**

RNA-seq analysis of CD4$^+$ T cells isolated from 3 WT and 3 *Stim1*$^{fl/fl}$*Cd4Cre* mice that were differentiated into pathogenic Th17 cells (pTh17: IL-1β, IL-6, IL-23) and non-pathogenic Th17 cells (npTh17: IL-6, TGFβ1).

A  Differentially expressed genes (DEG) in WT and STIM1-deficient npTh17 and pTh17 cells. $P_{adj.} < 0.10$.
B  Volcano plots of DEGs in npTh17 cells (gray dots). Highlighted in red (upregulated in *Stim1*$^{fl/fl}$*Cd4Cre*) and blue (downregulated in *Stim1*$^{fl/fl}$*Cd4Cre*) are DEGs that belong to a gene expression signature of npTh17 cells and pTh17 cells defined previously (Lee *et al*, 2012; Gaublomme *et al*, 2015).
C  Normalized expression of Th17 cell-associated cytokines and receptors in npTh17 cells derived from WT and *Stim1*$^{fl/fl}$*Cd4Cre* mice. Bar graphs represent the mean ± SEM from 3 mice per group. Statistical analysis by unpaired Student's *t*-test with the following significance levels: *$P < 0.05$, ***$P < 0.001$.
D  Summary of KEGG pathway analysis of npTh17 and pTh17 cells. Venn diagram shows pathways that are dysregulated in one or both Th17 subsets. $P_{adj.} < 0.05$ for all pathways. Some unique and shared pathways are indicated.
E  KEGG network analysis of npTh17 cells. $P_{adj.} < 0.05$ for all pathways.
F  Heat map of DEGs ($P_{adj.} < 0.1$) within the KEGG pathways "Foxo signaling" and "HIF-1 signaling" that are specifically dysregulated in npTh17 cells. Numerical values are relative gene expression.
G  Gene set enrichment analysis (GSEA) of DEGs in WT and STIM1-deficient npTh17 cells. Normalized enrichment score (NES) and *P*-values are as indicated for the following genesets: Myc targets: "Menssen_myc_targets"; Glycolysis: "Humancyc_mm_glycolysis_i"; Mitochondrial function: "Wong_mitochondria_gene_module"; TCA cycle: "Kegg_mm_citrate_cycle".

Data information: Statistical analysis in (A, F) by Wald *t*-test and correction using the Benjamini–Hochberg method; in (D, E) by hypergeometric distribution and correction using the Benjamini–Hochberg method; in (G) by phenotype-based permutation test.

and H). A similar defect in OCR was observed when HD CD4$^+$ T cells were treated with FK506 indicating that STIM1 regulates mitochondrial function in a calcineurin-dependent manner. It is noteworthy that similar metabolic defects in glycolysis and OXPHOS were not observed in anti-CD3/CD28 stimulated T cells of P1 and P2 after they had been cultured *in vitro* for several weeks (Appendix Fig S5A–D), which is consistent with their normal proliferation (Appendix Fig S2). These findings suggest that nutrient-rich and cytokine-supplemented cell culture conditions can overcome the metabolic defect of SOCE-deficient T cells or that SOCE is critical specifically for the metabolic adaptation of naive T cells after initial TCR stimulation. We had recently reported that combined deletion of STIM1 and STIM2 in murine T cells results in impaired activation of the PI3K-AKT-mTOR nutrient-sensing pathway (Vaeth *et al*, 2017a) accounting, at least partially, for their impaired glycolytic function. To investigate mTOR signaling in STIM1-deficient T cells, we stimulated PBMC of P1, P2, their mother, and a HD with anti-CD3/CD28 for 24 h and measured phosphorylation of mTOR (on S2448) and the ribosomal protein the p70S6 (on S235/236). CD4$^+$ and CD8$^+$ T cells of P1 and P2 had a marked defect in mTOR and p70S6 phosphorylation compared to T cells from their mother and a HD (Fig EV4). A comparable decrease in mTOR and p70S6 phosphorylation was observed after FK506 treatment of HD T cells, suggesting that mTOR signaling is regulated by SOCE and calcineurin signaling in *ex vivo* isolated T cells. Consistent with their normal proliferation and metabolic function, *in vitro* cultured T cells of P1 and P2 had no defect in mTOR and p70S6 phosphorylation (Appendix Fig S5E and F). Taken together, our data demonstrate that STIM1 plays a conserved role in the glycolytic function and mitochondrial respiration of human and mouse T cells.

## Discussion

We here report a novel LOF missense mutation in *STIM1* that abolishes SOCE by interfering with STIM1 function, but not its expression. The mutant STIM1 p.L374P protein is constitutively located near the PM and partially colocalizes with ORAI1 even before depletion of ER Ca$^{2+}$ stores, suggesting it is in a partially active state. The localization of mutant STIM1 at the PM does not depend on its PB

or expression of ORAI1, although it is able to bind to ORAI1 as detected by co-immunoprecipitation. Following store depletion, mutant STIM1 does not form puncta and its co-localization with ORAI1 does not increase unlike that of WT STIM1. Together these findings indicate that mutant STIM1 p.L374P protein retains its ability to bind to ORAI1 but fails to activate it, resulting in impaired SOCE. One explanation for the impaired activation of ORAI1 channels is that this process requires the formation of STIM1 clusters (Luik *et al*, 2008). Although mutant STIM1 is located near the PM, where it partially colocalizes with and binds to ORAI1, it fails to form puncta after store depletion, suggesting that the L374P mutation interferes with proper oligomerization of STIM1, which is required for the formation of macromolecular complexes with the correct STIM1:ORAI1 stoichiometry needed for CRAC channel activation (Yen & Lewis, 2019). An additional explanation for impaired SOCE could be that the L374P mutation results in conformational changes that do not abrogate STIM1 binding to ORAI1 but specifically interfere with its ability to gate the CRAC channel. This interpretation is supported by the finding that the binding of mutant STIM1 to ORAI1 observed in co-IP experiments is lost after cell stimulation, likely because it is replaced by active endogenous WT STIM1 that has a higher affinity for ORAI1. This changed or weaker binding implies that the interaction of mutant STIM1 with ORAI1 may be different in a fundamental way and prevents proper ORAI1 gating.

This behavior of STIM1 p.L374P is similar to that of another LOF mutation, STIM1 p.R429C, we had reported earlier (Maus *et al*, 2015). L374 and R429 are located in the second (CC2) and third (CC3) coiled-coil domains, respectively, within the cytosolic region of STIM1, which together form the cytoplasmic CAD (SOAR or CCb9) domain of STIM1 that is required for ORAI1 binding and CRAC channel activation (Kawasaki *et al*, 2009; Park *et al*, 2009; Yuan *et al*, 2009). Different models of STIM1-ORAI1 interaction exist based on available crystal and NMR structures (Yang *et al*, 2012; Stathopulos *et al*, 2013), and there are several potential explanations why the STIM1 p.L374P mutation interferes with SOCE. The NMR structure of a STIM1 (aa 312–387) – ORAI1 (aa 272–292) dimer complex (Stathopulos *et al*, 2013) shows that L374 is centrally located near each STIM-ORAI interaction site with hydrophobic residues that directly contact ORAI1-C$_{272–292}$ located

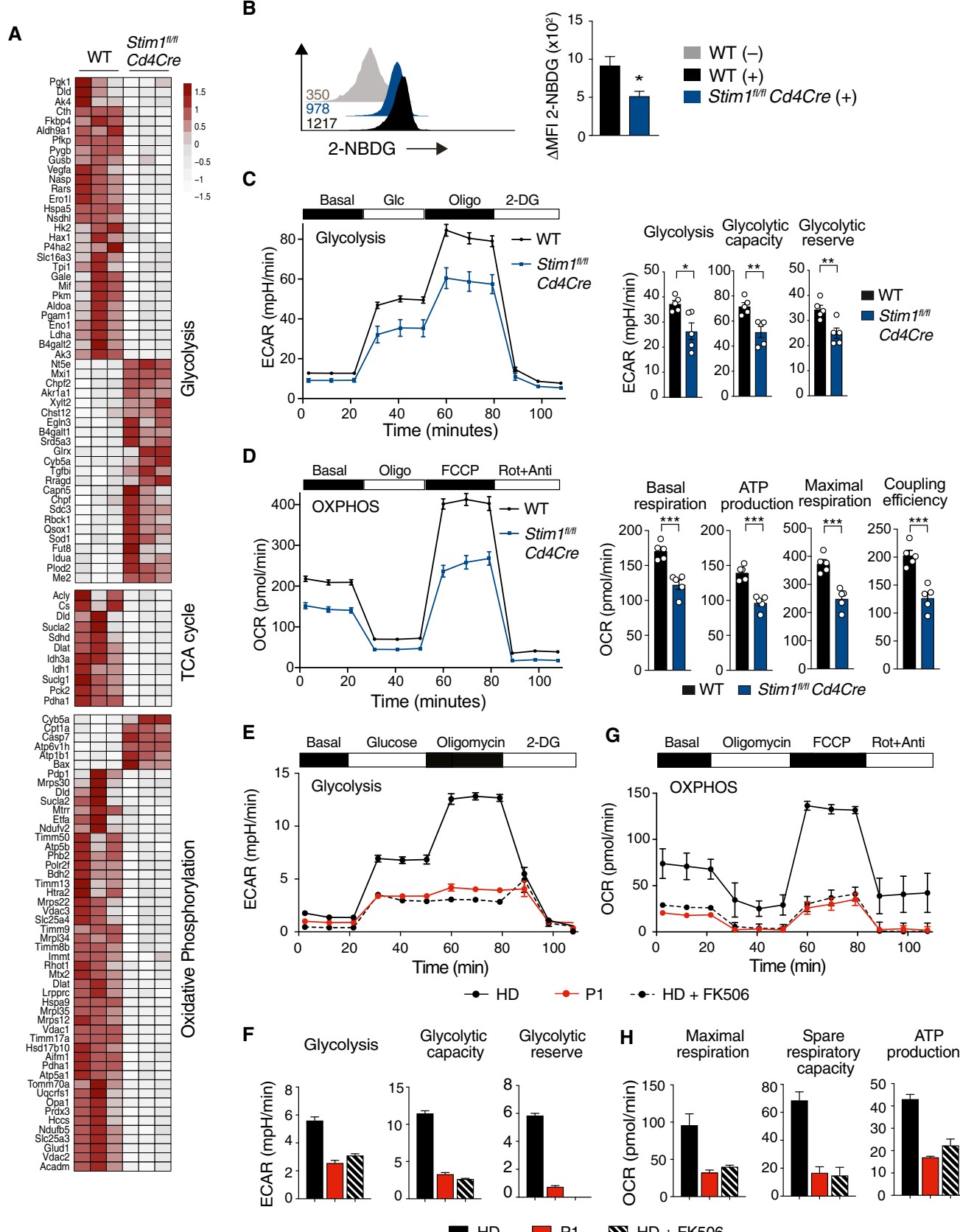

**Figure 8.**

**Figure 8. STIM1 controls aerobic glycolysis and oxidative phosphorylation in non-pathogenic murine Th17 cells and human T cells.**

A   Heat map of metabolism-associated DEGs (p (adj.) < 0.10) in npTh17 cells of WT and *Stim1*$^{fl/fl}$*Cd4Cre* mice determined by RNA-Seq as described in Fig 7. Heatmap shows relative minimum and maximum values per gene (row min/max). Numerical values are relative gene expression. TCA, tricarboxylic acid.

B–D   Glucose and mitochondrial metabolism in npTh17 cells from WT (gray or black) and *Stim1*$^{fl/fl}$*Cd4Cre* (blue)mice differentiated for 3 days *in vitro*. (B) Glucose uptake by npTh17 cells was measured using 2-NBDG (+) or not (−) and analyzed by flow cytometry 90 min later. Bar graphs show the delta MFI of 2-NBDG fluorescence normalized to unlabeled cells. Data represent the mean ± SEM from three independent experiments. (C) Extracellular acidification rate (ECAR) of npTh17 cells before and after addition of 25 mM glucose, 5 μM oligomycin, and 100 mM 2-deoxy-ᴅ-glucose (2-DG). Seahorse graphs and bar graphs depicting glycolysis, glycolytic capacity and glycolytic reserve represent the mean ± SEM from 2 mice per genotype in technical replicates from one representative out of three independent experiments with a total of 6 mice per genotype. (D) Oxygen consumption rate (OCR) of npTh17 cells before and after addition of 1 μM oligomycin, 1.5 μM FCCP, and 100 nM rotenone/1 μM antimycin. Seahorse graphs and bar graphs depicting basal and maximal respiration; coupling efficiency and ATP production represent the mean ± SEM from 2 mice per genotype in technical replicates from one representative out of three independent experiments with a total of 6 mice per genotype.

E–H   Glucose and mitochondrial metabolism in human CD4$^+$ T cells isolated from PBMC of a HD and P1 and stimulated with 5 μg/ml anti-CD3 and 10 μg/ml anti-CD28 for 24 h before analysis. Some HD T cells were stimulated in the presence of 1 μM FK506. (E,F) ECAR before and after addition of 25 mM glucose, 5 μM oligomycin, and 100 mM 2-DG. Seahorse graphs (E) and bar graphs (F) represent the mean ± SEM of 3 (HD), 4 (P1), and 2 (HD + FK506) technical replicates from isolated T cells from one experiment. (G,H) OCR before and after addition of 1 μM oligomycin, 1.5 μM FCCP, and 100 nM rotenone/1 μM antimycin. Seahorse graphs (G) and bar graphs (H) depicting basal and maximal respiration, SRC and ATP production represent the mean ± SEM of 2 technical replicates from isolated T cells from one experiment.

Data information: Statistical analysis in (A) by Wald *t*-test and correction using the Benjamini–Hochberg method; in (B–D) by unpaired Student's *t*-test with the following significance levels: *$P < 0.05$, **$P < 0.01$, ***$P < 0.001$.

N- and C-terminally to L374. Our NMR-based model suggests that loss of the free backbone NH due to the L374P substitution results in an altered hydrogen bond network near the mutation, which perturbs the structure and side-chain orientations in the immediate vicinity of L374P that are important for STIM1-ORAI1 interaction.

The crystal structure of the human STIM1 CAD dimer has a "V"-shaped structure (Yang *et al*, 2012) with two apex regions critical for interactions with ORAI1 (Yang *et al*, 2012; Zhou *et al*, 2015). Our homology model of the M374P mutation in the context of the CAD crystal structure shows a movement of each apex position by ~6.4 Å compared to the template dimer (note that the CAD template is a L374M/V419A/C437T triple mutant engineered to promote crystallization and yield a high-resolution structure (Yang *et al*, 2012)). This apex movement is caused by the perturbation of the backbone hydrogen bond network and a limited backbone rotation around proline. These structural changes increase the distance between the apex regions within dimers (a ~5.0 Å increase in the F394-F394′ intermolecular distance), which is predicted to interfere with STIM1 binding to ORAI1 in a proposed bimolecular STIM-ORAI interaction model in which each STIM1 dimer subunit interacts with one ORAI1 subunit of the same hexameric CRAC channel complex (Stathopulos *et al*, 2013). Interestingly, M374P in CC2 is located very near the intramolecular CC3 interface in the CAD crystal structure (Yang *et al*, 2012). Given the altered hydrogen bonding around M374P, the position of residue L373 is likely perturbed due to the L374P mutation, potentially triggering a conformational extension in the cytosolic CC domains of STIM1, which is consistent with the finding that a L373S mutation causes a similar conformational extension of STIM1 (Frischauf *et al*, 2009; Zhang *et al*, 2013). Together, our data support a model in which L374 is required to keep STIM1 in a resting state and regulate its proper binding to ORAI1.

Increased susceptibility to fungal infections in humans is caused by a wide range of defects in innate and adaptive immunity. SCID and CID due to defects in T-cell development and function, respectively, are associated with fungal infections including chronic mucocutaneous candidiasis (CMC) and *Pneumocystis jirovecii* pneumonia (Lanternier *et al*, 2013). Defects in IL-17 signaling and Th17 cell function cause CMC, which may result from inherited defects in IL-17RA, IL-17RC, and IL-17F genes, LOF mutations in STAT3, or GOF

mutations in STAT1 (Hernandez-Santos & Gaffen, 2012). Mutations in AIRE resulting in autoimmune polyglandular syndrome type 1 (APS1) are associated with the formation of autoantibodies against IL-17A, IL-17F, and IL-22 and CMC (Lanternier *et al*, 2013; Mengesha & Conti, 2017). Patients with CRAC channelopathy due to LOF mutations in *ORAI1* and *STIM1* suffer from life-threatening viral and bacterial infections early in life (Feske, 2010; Vaeth & Feske, 2018). Several patients have been reported to also suffer from fungal infections (McCarl *et al*, 2009; Badran *et al*, 2016; Klemann *et al*, 2017; Lian *et al*, 2018). Most of them had mucosal or cutaneous *C. albicans* infections, and at least one had gastrointestinal candidiasis, whereas others suffered from infections with *Aspergillus fumigatus* or *Pneumocystis jirovecii*. The most severe form of fungal infection associated with CRAC channelopathy reported to date was a systemic *C. albicans* infection in a patient with *ORAI1* p.L194P LOF mutation (Lian *et al*, 2018). The patients with STIM1 p.L374P mutation reported here presented with severe onychomycosis. Since most fungal infections in CRAC channel-deficient patients are not life-threatening, it is possible that their incidence is underreported. The underlying causes of increased susceptibility to fungal infection in these patients likely involve a combination of defects in innate and adaptive immunity. We here show that SOCE is required for immunity to mucosal infection with *C. albicans* infection as deletion of STIM1 or both STIM1 and STIM2 in all immune cells (including T cells, ILCs, neutrophils, and macrophages) strongly exacerbated the severity of mucosal candidiasis. Abolishing SOCE compromised the ability of neutrophils to kill ingested *C. albicans* and to produce ROS upon PKC stimulation, likely contributing to impaired immunity to mucosal fungal infection.

By contrast, T cell-specific deletion of STIM1 did not result in increased susceptibility to mucosal *Candida* infection. This finding is in line with the observation that complete deletion of T and B cells in *Rag1*$^{-/-}$ mice results in only moderately increased *C. albicans* burdens at day 7 p.i. (Gladiator *et al*, 2013; Conti *et al*, 2014). In humans, by contrast, T cells play an important role in adaptive immunity to mucosal candidiasis as evidenced by the high susceptibility of AIDS and T cell-deficient patients to *C. albicans* infections. We here show that SOCE is essential for the function of *Candida*-primed human T cells. T cells of HDs cultured in the presence of

*C. albicans* produced robust amounts of IFN-γ, TNF-α, and IL-17A, cytokines that are important for immunity to fungal infection (Conti & Gaffen, 2015; Netea *et al*, 2015). Cytokine expression was abolished, however, in HD T cells treated with a CRAC channel inhibitor and in T cells of patients with the STIM1 p.L374P mutation. The limited role of murine T cells in immunity to mucosal candidiasis may be due to the fact that mice housed under SPF conditions are immunologically naive to *C. albicans* and infection causes an acute immune response (Conti *et al*, 2014). Nevertheless, we found that SOCE in murine T cells is critical for immunity to systemic fungal infection. T cell-specific deletion of STIM1 resulted in increased weight loss, fungal dissemination, and mortality after systemic *C. albicans* infection of *Stim1*^fl/fl^*Cd4Cre* mice compared to WT littermates, which was associated with reduced production of IFN-γ, TNF-α, and IL-17A in the absence of STIM1. These findings are consistent with the important role of Th1 and Th17 cells for immunity to systemic fungal infection (Netea *et al*, 2003; Huang *et al*, 2004; van de Veerdonk *et al*, 2010; Shao *et al*, 2019). Furthermore, the increased kidney inflammation of *Stim1*^fl/fl^*Cd4Cre* mice we observed after systemic *C. albicans* infection is consistent with a similar kidney pathology due to conditional deletion of IL-17RA in renal tubular epithelial cells (RTEC) (Ramani *et al*, 2018), indicating that IL-17 signaling is critical to prevent kidney inflammation and damage after systemic *C. albicans* infection.

Th17 cells are essential for immunity to fungal pathogens and extracellular bacteria (Curtis & Way, 2009). Pathogenic Th17 cells, on the other hand, promote inflammation in a variety of autoimmune disorders such as MS, colitis, rheumatoid arthritis (RA), and psoriasis (Stockinger & Omenetti, 2017) emphasizing the dual role of Th17 cells in immunity. We previously showed that SOCE is essential for pathogenic Th17 cell function as mice with T cell-specific deletion of ORAI1, STIM1 or STIM2 were protected in Th17 cell-dependent animal models of autoimmune diseases such as EAE and colitis (Ma *et al*, 2010; McCarl *et al*, 2010; Schuhmann *et al*, 2010; Kaufmann *et al*, 2016). In another model of pathogenic Th17 cell-driven disease, caused by expression of a hyperactive form of STAT3, we demonstrated that STIM1 and SOCE are required to cause pulmonary and skin inflammation (Kaufmann *et al*, 2019). At the molecular level, STIM1 was found to regulate the expression of many genes required for mitochondrial function, especially OXPHOS.

We here show that SOCE is also critical for the function of non-pathogenic Th17 cells during *C. albicans* infection and that SOCE has markedly different effects on gene expression and the molecular function of non-pathogenic and pathogenic Th17 cells. In the absence of STIM1, the majority of genes associated with a non-pathogenic Th17 cell signature (Lee *et al*, 2012; Gaublomme *et al*, 2015) were upregulated in Th17 cells of *Stim1*^fl/fl^*Cd4Cre* mice. By contrast, genes associated with a pathogenic Th17 cell signature were equally up- and downregulated in the absence of STIM1. These findings suggest that STIM1-deficient non-pathogenic Th17 cells largely maintain their molecular identity with the exception of cytokine production (IL-17A, IL-17F), whereas deletion of STIM1 results in a greater loss of genes previously shown to define pathogenic Th17 cells.

We furthermore found that lack of STIM1 has differential effects on key metabolic pathways in non-pathogenic and pathogenic Th17 cells. STIM1-deficient non-pathogenic Th17 cells had impaired expression of many glycolysis-associated genes, which was not observed in pathogenic Th17 cells. A likely mechanistic explanation

for this finding is that many genes associated with HIF-1α and myc signaling were downregulated in STIM1-deficient non-pathogenic Th17 cells, whereas most genes in the Foxo signaling pathway including Foxo1 itself were upregulated. HIF-1α and myc positively regulate several key glycolytic enzymes, whereas Foxo1 negatively regulates myc expression (Buck *et al*, 2017; Newton *et al*, 2018). Accordingly, we observed significantly reduced glucose uptake and glycolysis in STIM1-deficient non-pathogenic Th17 cells. These findings are consistent with the critical role of SOCE in regulating the expression of glucose transporters and glycolytic enzymes by controlling HIF-1α, myc, and IRF4 levels in naive T cells after TCR stimulation (Vaeth *et al*, 2017a). Importantly, the dependence of glycolysis on STIM1 was specific to non-pathogenic Th17 cells and not observed in pathogenic Th17 cells. Normal glycolysis in STIM1-deficient pathogenic Th17 cells is in line with our previous study of STAT3-dependent pathogenic Th17 cells (Kaufmann, *et al*, 2019). Besides glycolysis, STIM1 also regulates the expression of TCA cycle enzymes and mitochondrial ETC complexes in non-pathogenic Th17 cells. Accordingly, OXPHOS was markedly reduced in these cells. The role of SOCE in mitochondrial function is shared between non-pathogenic and pathogenic Th17 cells. This finding is consistent with the fact that deletion of STIM1 in STAT3-dependent pathogenic Th17 cells abolishes expression of many mitochondrial ETC genes and impairs OXPHOS (Kaufmann *et al*, 2019). Collectively our data demonstrate that SOCE is essential for mitochondrial function and OXPHOS in both pathogenic and non-pathogenic Th17 cells, whereas its role in glycolysis is specific to non-pathogenic Th17 cells. Similar defects in glycolysis and mitochondrial respiration were observed in T cells of the *STIM1* p.L374P mutant patients.

Taken together, we here show that STIM1 is a critical regulator of antifungal immunity by regulating the expression of cytokines and metabolic pathways in non-pathogenic Th17 cells, which may explain the increased susceptibility of patients with *STIM1* and *ORAI1* mutations to fungal infections. It is intriguing to speculate that the different roles of SOCE in the metabolic regulation of non-pathogenic Th17 cells (OXPHOS and glycolysis) and pathogenic Th17 cells (OXPHOS, but not glycolysis) may potentially be exploited for the treatment of Th17-cell-mediated autoimmune diseases such as MS or rheumatoid arthritis without compromising antifungal immunity.

# Materials and Methods

### Patients

Detailed case reports of P1 and P2 are provided in Supplemental Information. Written informed consent was obtained from both patients, in accordance with research ethics board policies at the University of British Columbia and Oregon Health & Science University. Experiments using deidentified cell samples of patients were conducted with Institutional Review Board approval at the New York University School of Medicine.

### Mice

The generation of *Stim1*^fl/fl^ and *Stim1*^fl/fl^*Stim2*^fl/fl^ has been described before (Oh-Hora *et al*, 2008). These mice were further crossed to

Cd4-Cre mice (B6.Cg-Tg(Cd4-cre)1Cwi/BfluJ, JAX strain 22071), Vav-iCre mice (B6.Cg-Commd10Tg(Vav1-icre)A2Kio/J, JAX strain 008610), and Mx1Cre mice (B6.Cg-Tg(Mx1-cre)1Cgn/J, JAX strain 003556) as described (Vaeth *et al*, 2015; Saint Fleur-Lominy *et al*, 2018). To induce deletion of *Stim1* and *Stim2* genes in hematopoietic lineage cells, *Stim1^{fl/fl}Mx1Cre* and *Stim1^{fl/fl}Stim2^{fl/fl}Mx1Cre* mice were injected i.p. with 10 μg/g body weight of poly(I:C) (Sigma-Aldrich) every other day for 1 week followed by 1 month without treatment. Mice from both sexes were evenly used for all experiments and were between 8 and 14 weeks old at the time of analysis. All mice are on the C57BL/6 genetic background. Animals were maintained under SPF conditions in accordance with institutional guidelines for animal welfare approved by the IACUC at NYU School of Medicine.

## Cell lines

Flip-in T-Rex HEK293 cells with inducible GFP-ORAI1 expression were a gift of T. Shuttleworth (University of Rochester, Rochester, NY). Cells were maintained in DMEM (Fisher) supplemented with 10% FCS, 2 mM L-glutamine, 100 U/ml, and 0.1 mg/ml streptomycin (Sigma-Aldrich) at 37°C, 10% $CO_2$. GFP-ORAI1 expression was induced by addition of 1 μg/ml doxycycline (Sigma-Aldrich) for 24 h as described previously (Thompson & Shuttleworth, 2012; Maus *et al*, 2015).

## DNA sequencing analysis

Genomic DNA was extracted from PBMC using the Flexigene DNA Kit (QIAGEN). PCR was conducted using primers flanking all 12 exons of the *STIM1* gene. Primers used to amplify and sequence exon 8 were as follows: fwd5′ AAAGCAGATAAGAAGTCT-GAGTTCTG and rev5′ ACCACCAGGATATCTCTTCAC; PCR products were separated on 1.5% agarose gel, excised using the QIAGEN gel extraction kit and sequenced directly (Macrogen Inc., NY) using the same PCR primers.

## Genetic analysis

Variant analysis was performed using the Genome Aggregation Database (gnomAD) https://gnomad.broadinstitute.org. To predict the deleteriousness of the p.L374P mutation, the Combined Annotation-Dependent Depletion (CADD) score was obtained using https://cadd.gs.washington.edu/snv and the CADD model GRCh37-v1.4. The genomic position of the mutation was determined using https://mutalyzer.nl and human genome assembly GRCh37 (hg19).

## Homology modeling of STIM1 p.L374P mutant

Twenty homology models of M374P CAD/SOAR/CCb9 were generated using 3TEQ.pdb (which harbors the L374M/V419A/C437T triple mutation) as the template structure in MODELLER (version 9.16) (Webb and Sali 2014) to model the structural changes caused by the human p.L374P mutation. Similarly, 20 homology models of the human p.L374P CC1_{TM-distal}-CC2:ORAI1-C272-292 complex were generated using the lowest energy structure of 2MAK.pdb as the template structure in MODELLER. The models with the lowest DOPE score were taken for visualization and structural analyses. These

models orient the mutant side chains in positions that are homologous to the template residues. All structure figures, distance measurements, and backbone hydrogen bond identifications were done using PyMOL (Version 1.7.4, Schrödinger, LLC).

## Intracellular $Ca^{2+}$ measurements

Fibroblasts, T cells, and PBMC were labeled with 2 μM Fura-2 AM (Life Technologies) for 30 min in RPMI 1640 medium supplemented with 10% FCS, 1% L-glutamine, and 1% penicillin/streptomycin. Cells were attached for 10 min to 96-well glass-bottom plates (Fisher) that had been precoated with 0.01% poly-L-lysine (w/v) (Sigma-Aldrich) for 2 h. Intracellular $Ca^{2+}$ measurements were performed using a Flexstation 3 plate reader (Molecular Devices). Cells were stimulated with 1 μM thapsigargin (TG) in $Ca^{2+}$-free Ringer solution (155 mM NaCl, 4.5 mM KCl, 3 mM $MgCl_2$, 10 mM D-glucose, 5 mM Na HEPES), followed by addition of 2 mM $Ca^{2+}$ Ringer solution (final $[Ca^{2+}]_o$ 1 mM) to induce SOCE. Fura-2 emission ratios (F340/380) were acquired at 510 nm following excitation at 340 and 380 nm every 5 s. F340/380 ratios were quantified by analyzing the integrated $Ca^{2+}$ signal (area under the curve, AUC) after TG stimulation or readdition of extracellular $Ca^{2+}$ and by analyzing the peak F340/380 response after $Ca^{2+}$ readdition (normalized to the baseline F340/380 ratio before $Ca^{2+}$ readdition) using GraphPad Prism 8.0 software.

## Total internal reflection microscopy

Flip-in T-Rex HEK293 cells stably transfected with doxycycline-inducible GFP-ORAI1 were plated onto UV-sterilized coverslips and transfected with mCherry-STIM1 (WT or the following mutants: p.L374P, ΔK lacking the C-terminal PB, p.L374P-ΔK). One day after STIM1 transfection, GFP-ORAI1 expression was induced by 1 μg/ml doxycycline. ORAI1 expression on the surface of HEK293 cells was confirmed by flow cytometry using a mouse anti-human ORAI1 monoclonal IgG1 antibody (1:200), clone 29A2, that recognizes an epitope (amino acids 196–234) in the second extracellular loop of ORAI1 (Vaeth *et al*, 2017b). A mouse IgG1 was used as isotype control and a goat anti-mouse IgG-Alexa Fluor 647 as secondary antibody (Life Technologies, 1:500). TIRFM images for Fig 2 were acquired using a Nikon Eclipse Ti inverted microscope, an Apo TIRF 100×/NA1.49 oil-immersion DIC objective, and an Andor iXon DU897 electron-multiplier charge-coupled device (EMCCD); two laser lines (488 and 568 nm; Coherent) were used to acquire fluorescent images. TIRFM images for Fig EV1 were acquired using an Olympus IX81 inverted epifluorescence microscope, a 60×/NA1.45 oil-immersion objective, and EMCCD (Hamamatsu); two laser lines (488 and 543 nm; Melles Griot) were used sequentially to acquire fluorescent images. Experiments were performed at 37°C (Fig 2) or room temperature (Fig EV1). Image analysis was conducted using NIH ImageJ Fiji. Recruitment of mCherry-STIM1 to ER–PM junctions in TIRFM images was evaluated by calculating the mCherry fluorescence over time [F(t)] after ER store depletion. The mCherry signal was either normalized to baseline mCherry fluorescence (F0) image for each individual cell using NIH ImageJ Fiji or subtracted as mean fluorescent intensity (MFI) using NIH ImageJ Fiji. The co-localization of GFP-ORAI1 with mCherry-STIM1 in the TIRFM evanescent field was measured

by using the co-localization analysis plug-in from NIH ImageJ Fiji. Pearson's coefficients were calculated for each time point and cell individually using NIH ImageJ Fiji.

## Western blotting

Expanded human T cells were lysed in RIPA buffer (10 mM Tris–HCl, pH 7.5, with 150 mM NaCl, 1% Triton X-100, 1% Nonidet P-40, 2 mM EDTA, 0.2 mM phenylmethylsulfonyl fluoride, and protease inhibitor mixture), and total protein extracts (30–50 μg) were resolved by 4–20% SDS–PAGE. Proteins were transferred onto nitrocellulose membranes (Bio-Rad), incubated with a polyclonal rabbit anti-human antibody to STIM1 (1:1,000) described earlier (Picard *et al*, 2009) or a polyclonal antibody to β-actin (Santa Cruz,1:500). Fluorescently labeled secondary antibodies (LICOR) were used for protein detection using a Li-Cor Odyssey Fc Image system (Li-Cor).

## Immunoprecipitation

HEK293 cells were co-transfected with Flag-ORAI1 and mCherry-STIM1 (WT or mutant p.L374P). One day after transfection, cells were treated with either 2 mM $Ca^{2+}$ Ringer solution or 1 μM Thapsigargin in $Ca^{2+}$ free Ringer solution. Cells were lysed in RIPA buffer as described above. Cells lysates were incubated with anti-Flag (clone M2, Sigma-Aldrich, 5 μg/ml) overnight at 4°C. Protein complexes were purified on protein A/G-agarose beads (Pierce) and eluted by competition with Flag peptide (0.4 μg/μl) for 1 h at 4°C. Immunoprecipitated proteins and 10% of total protein extract (input control) were separated by 4–20% SDS–PAGE. Proteins were detected by immunoblotting using either HRP coupled anti-Flag antibody (Sigma-Aldrich, 1:2,000), or a polyclonal rabbit-anti-human antibody to STIM1 (1:1,000) described previously (Picard *et al*, 2009) or a monoclonal mouse anti-human antibody to β-actin (Proteintech, 1:5,000) followed by an HRP coupled secondary antibody (Sigma-Aldrich, 1:7,500). Signals were detected using Western Lightning Plus-ECL (PerkinElmer) with the Amersham Imager 680 (GE Healthcare). Image analysis was performed by densitometry using NIH ImageJ software.

## PBMC isolation and human T-cell culture

PBMCs were isolated from whole blood samples by density centrifugation using Ficoll-Paque plus (GE Amersham). Human CD4$^+$ T cells isolation was performed using CD4 MicroBeads (Miltenyi Biotec). PBMCs or purified CD4$^+$ T cells were stimulated with 5 μg/ml plate-bound anti-CD3 (clone OKT3) and 10 μg/ml soluble anti-CD28 (clone CD28.2, both eBioscience) monoclonal antibodies in the presence or absence of 1 μM FK506 (Sigma-Aldrich) in RPMI 1640 medium supplemented with 10% FCS, 1% L-glutamine, and 1% penicillin/streptomycin (Mediatech Inc., VA). For the expansion of human CD4$^+$ T cells, PBMC were stimulated with 1 μg/ml PHA (Thermo Fisher Scientific), irradiated buffy coat cells, and irradiated EBV-transformed B cells. Cells were cultured in RPMI 1640 medium supplemented with 2% human serum and 10% FCS, 1% L-glutamine, 1% penicillin/streptomycin, 25 mM HEPES, and 50 U/ml recombinant human IL-2.

## Murine CD4$^+$ T-cell isolation and Th17 cell differentiation

CD4$^+$ T cells were isolated from the spleen and submandibular, axillar, inguinal, and mesenterial lymph nodes of mice by negative enrichment using a MagniSort mouse CD4$^+$ T-cell enrichment kit (eBioscience) and stimulated with 0.25 μg/ml plate-bound anti-CD3 (clone 2C11) and 1 μg/ml anti-CD28 (clone 37.51; both Bio X Cell) antibodies in the presence of 2 μg/ml anti-IL-4 (clone 11B11) and 2 μg/ml anti-IFN-γ (clone XMG1.2; both eBioscience). For the differentiation into non-pathogenic Th17 cells, CD4$^+$ T cells were cultured in the presence of 20 ng/ml IL-6 (Peprotech) and 0.5 ng/ml hTGFβ1 (PeproTech) for 2 or 3 days. For the differentiation into pathogenic Th17 cells, CD4$^+$ T cells were cultured in the presence of 20 ng/ml IL-6 (Peprotech), 20 ng/ml IL-1β (Peprotech), and 20 ng/ml IL-23 (eBioscience) for 2 or 3 days.

## T-cell proliferation

PBMC or expanded human T cells were loaded with 2.5 μM CFSE (Molecular Probes) and stimulated with anti-CD3 and anti-CD28 antibodies in the presence or absence of 1 μM FK506. After 96 h, cells were stained for CD3, CD4, and CD8 and analyzed by flow cytometry for surface markers, cell size (FSC), and CFSE dilution using a LSRII flow cytometer (BD Biosciences) and FlowJo v.8.7 software (Tree Star).

## cDNA synthesis and quantitative real-time PCR

Total RNA was extracted from human PBMC using Trizol Reagent (Invitrogen, Carlsbad, California) as described before *(*McCarl *et al*, 2009*)*. cDNA was synthesized using Superscript II™ RT (Invitrogen, Carlsbad, CA) according to the manufacturer's instructions. PCRs were performed using the iCycler system (Bio-Rad Laboratories) and SYBR green PCR kit (Applied Biosystems, Foster City, CA). Primers used: STIM1 (forward: ACACAGGGGCTTGTCAATTC; reverse: GTCACAGTGAGAAGGCGACA) and STIM2 (forward: GCATGGTGGACTCAGTGACA; reverse: ACTGGCTCTGCCGCAACT).

## Intracellular cytokine staining by flow cytometry

For intracellular cytokine measurements, total splenocytes (systemic infection) or total lymph node cells (sublingual infection) were isolated and restimulated with 20 nM phorbol 12-myristate 13-acetate (PMA, Calbiochem) and 1 μM ionomycin (Invitrogen) in the presence of 5 μM brefeldin A for 4–6 h followed by washing of the cells in PBS containing 2% FBS and incubation with anti-FcγRII/FcγRIII antibodies (2.4G2, eBioscience) for 20 min. Cell surface staining with fluorescently labeled antibodies was performed at room temperature for 30 min in the dark followed by permeabilization and fixation by incubation of the cells with IC Fix Buffer (eBioscience) for 30 min. Thereafter, cells were washed and stained with antibodies against cytokines. A complete list of anti-human and anti-mouse antibodies used in this study can be found in Appendix Table S2. For all flow cytometry experiments, cells were analyzed using a LSRII flow cytometer (BD Biosciences) and FlowJo v.8.7 software (Tree Star).

**Analysis of mTOR signaling and STIM1 protein expression by flow cytometry**

PBMC were stimulated for 24 h with anti-CD3/CD28 in the presence or absence of 1 μM FK506. After antibody staining of CD4 and CD8 at the cell surface, fixation, and permeabilization, the phosphorylation of proteins in the mTOR signaling pathway was detected by using anti-phospho-mTOR Ser2448 (MRRBY) and anti-phospho-S6 Ser235/236 (cupk43k, all eBioscience) antibodies. For the analysis of STIM1 protein expression, expanded human T cells were fixed and permeabilized with IC Fix Buffer (eBioscience) followed by intracellular staining with an unconjugated, affinity-purified, polyclonal rabbit-anti-human antibody directed against a C-terminal STIM1 epitope that has been described earlier (Picard *et al*, 2009). Antibody binding to STIM1 was visualized using a secondary, Alexa Fluor 647-conjugated mouse anti-rabbit antibody. Expanded human T cells incubated with serum from non-immunized rabbits followed by treatment with secondary antibody served as negative control.

**_Candida albicans_ culture and CFU counts**

$1 \times 10^7$ cells of the *C. albicans* strain UC 820 (ATCC, MYA-3573) were added to YPD medium (Sigma-Aldrich), placed in a shaking incubator at 28°C for 24 h, centrifuged at 1,000 rpm for 5 min, and resuspended in fresh 15 ml YPD media followed by culture of *C. albicans* on YPD agar plates (Sigma-Aldrich) for another 24 h to determine CFUs before systemic or oral infection of mice.

**Sublingual and systemic candida infection**

Mice were anesthetized with a mixture of ketamine and xylazine before *C. albicans* infections. For systemic infection, $2 \times 10^5$ CFUs of *C. albicans* were injected i.v. into the retrobulbar venous plexus and the body weight monitored for the duration of the experiment. Mice were sacrificed at day 6 p.i. to analyze fungal burdens and tissue inflammation and to isolate cells. For sublingual *C. albicans* infection, mice were infected by placing cotton-tipped applicators saturated with $2 \times 10^7$ CFUs of *C. albicans* under the tongue for 90 min. Mice were sacrificed at day 7 p.i. to collect organs, which were homogenized using a Power Gen 125 tissue homogenizer (Fisher Scientific). Fungal burdens were measured by culturing tissue homogenates at different dilutions on YPD agar plates for 24 h at 28°C followed by determination of CFUs.

**Histology**

Tongues and kidneys of mice infected with *C. albicans* sublingually or *i.v.*, respectively, were formalin-fixed and paraffin-embedded (FFPE), cut into 5-μm slices, and stained with hematoxylin and eosin (H&E) or periodic acid–Schiff (PAS) in 95% ethanol using standard protocols. Images were acquired using a SCN400 slide scanner (Leica) and viewed with Slidepath Digital Image Hub (Leica). Leukocytes and fungal burdens were evaluated in H&E and PAS-stained slides, respectively.

**Candida killing assay**

CD11b$^+$Gr1$^+$ neutrophils from poly(I:C) treated WT, *Stim1$^{fl/fl}$ Mx1Cre,* and *Stim1$^{fl/fl}$Stim2$^{fl/fl}$Mx1Cre* mice were cocultured with

*C. albicans in vitro* for 0, 30, 60, and 90 min at a MOI 0.5 before neutrophils were lysed. Cell lysates were plated on YPD agar plates for 24 h to determine CFUs of live *C. albicans*.

**ROS production by neutrophils**

CD11b$^+$Gr1$^+$ neutrophils isolated from poly(I:C) treated WT, *Stim1$^{fl/fl}$Mx1Cre,* and *Stim1$^{fl/fl}$Stim2$^{fl/fl}$Mx1Cre* mice were incubated with 50 μM dihydrorhodamine 123 (DHR 123) for 10 min at 37°C. The non-fluorescent DHR 123 is oxidized to rhodamine 123 (R 123) in the presence of $H_2O_2$. After neutrophil stimulation with 10 nM PMA for 30 min, DHR 123 fluorescence was measured by flow cytometry.

**Metabolic analyses**

Oxygen consumption rates (OCR) and extracellular acidification rates (ECAR) were measured using an XFe24 Extracellular Flux Analyzer (Seahorse Bioscience). Before analysis, human CD4$^+$ T cells enriched from PBMC and murine Th17 cells differentiated from naive CD4$^+$ T cells were counted. OCR was measured using the mitochondria stress test protocol. Cells were resuspended in XF media (Seahorse Biosciences) supplemented with 10 mM glucose (Sigma-Aldrich), 1 mM GlutaMAX (GIBCO), and 1 mM sodium pyruvate (Corning). OCR was measured under basal conditions and upon treatment with the following reagents: the ATP synthase inhibitor oligomycin (1 μM); the protonophore carbonyl cyanide-4-(trifluoromethoxy)phenylhydrazone (FCCP) (1.5 μM) to uncouple mitochondria; the mitochondrial complex I inhibitor rotenone (100 nM); and the mitochondrial complex III inhibitor antimycin A (1 μM). For ECAR measurements, cells were resuspended in XF media (Seahorse Biosciences) supplemented with 1 mM GlutaMAX (GIBCO) and 1 mM sodium pyruvate (Corning). ECAR was measured under basal conditions and upon addition of the following reagents: glucose (25 mM); oligomycin (5 μM); and 2-DG (100 mM) to block glycolysis. Different OXPHOS and glycolysis parameters were calculated as described previously (Vaeth *et al*, 2017a).

**Glucose uptake**

Glucose uptake was measured by loading murine non-pathogenic Th17 cells with the fluorescent glucose analog 2-NBDG (Thermo Fisher Scientific). Stimulated and unstimulated Th17 cells were incubated in glucose-free RPMI medium containing 100 μM 2-NBDG for 90 min at 37°C, and the amount of 2-NBDG taken up by cells was measured by flow cytometry after surface staining for CD4 for 15 min at RT.

**RNA sequencing and data processing**

Total RNA was extracted from $1 \times 10^6$ non-pathogenic and pathogenic Th17 cells using the RNeasy Micro RNA Isolation Kit (Qiagen). The RNA quality and quantity were analyzed on a Bioanalyzer 2100 (Agilent) using a PICO chip and samples with an RNA integrity number (RIN) of > 9 were used for library preparation. RNA-seq libraries were prepared using the TruSeq RNA sample prep v2 kit (Illumina), starting from 100 ng of DNAse I (Qiagen)-treated total RNA, following the manufacturer's protocol with 15 PCR cycles.

## The paper explained

### Problem

Calcium signals are critical for the function of cells of the innate and adaptive immune system and their ability to mediate protective immune responses to infection. Calcium signals in immune cells are mediated by CRAC channels that are formed by ORAI1 and STIM1 proteins. We had previously reported that defects in this pathway render patients and mice susceptible to viral infections. The role of CRAC channels for immunity to infection with fungal pathogens has not been studied and the mechanisms by which calcium signals regulate antifungal immunity are largely unexplored.

### Results

We here describe patients with an inherited novel loss-of-function mutation in the *STIM1* gene that abolishes calcium influx through CRAC channels and therefore the function of immune cells. These patients, like others with mutations in the same pathway described before, are more susceptible to fungal infections with *C. albicans, A. fumigatus, P. jirovecii,* and potentially other fungal pathogens. In this study, we describe the molecular mechanisms by which the mutation abolishes the ability of STIM1 to activate CRAC channels and show that lack of calcium influx in the patients' T cells suppresses several metabolic pathways that are required for normal T-cell function. To understand the mechanisms by which CRAC channels control immunity to fungal infections, we used mice with genetic deletion of STIM1 and its homologue STIM2 to abolish calcium influx in all immune cells or more selectively only in T cells. Mice lacking STIM1 or both STIM1 and STIM2 in all immune cells showed increased susceptibility to oral *C. albicans* infection, which was associated with defective neutrophil function. Deletion of STIM1 only in T cells, by contrast, had little effect on immunity to oral *C. albicans* infection but rendered mice susceptible to systemic fungal infection. A subset of CD4$^+$ T cells, T helper (Th) 17 cells, are important mediators of antifungal immunity. Deletion of STIM1 in Th17 cells impaired not only the expression of several Th17 cytokines but also that of many genes which regulate the metabolic function of Th17 cells. This included genes controlling the utilization of glucose by aerobic glycolysis and the generation of ATP in mitochondria by oxidative phosphorylation (OXPHOS). In contrast to Th17 cells that mediate antifungal immunity, a related subset of Th17 cells that cause inflammation in the context of many autoimmune diseases required CRAC channel function only to regulate OXPHOS but not glycolysis.

### Impact

Our study offers new insights into the role of calcium influx through CRAC channels in cells of the innate and adaptive immune system and how this signaling pathway provides immunity to fungal pathogens. Furthermore, we describe distinct roles of CRAC channels in regulating the metabolic function of Th17 cell subsets that contribute to antifungal immunity and those that mediate inflammation in autoimmune diseases like multiple sclerosis, Crohn's disease, and rheumatoid arthritis. We propose that the latter finding may potentially be exploited for the treatment of Th17 cell-mediated autoimmune diseases.

The amplified libraries were purified using AMPure beads (Beckman Coulter), quantified by Qubit 2.0 fluorometer (Life Technologies), and visualized using an Agilent Tapestation 2200. The libraries were pooled equimolarly and loaded on the HiSeq 2500 Sequencing System (Illumina) and run as single 50-nucleotides reads generating about 30 million reads per sample. Sequencing reads were mapped to the mouse reference genome (GRCm38.85/mm10) using the STAR aligner (v2.5.0c; Dobin *et al*, 2013).

Alignments were guided by a Gene Transfer Format file (GTF GRCm38.85). The mean read insert sizes and their standard deviations were calculated using Picard tools (v.1.126) (http://broadinstitute.github.io/picard). The read count tables were generated using HTSeq (v0.6.0) (Anders *et al*, 2015), normalized based on their library size factors using DEseq2(Love *et al*, 2014), and differential expression analysis was performed. The Read Per Million (RPM) normalized BigWig files were generated using BEDTools (v2.17.0) (Quinlan & Hall, 2010) and bedGraphToBigWig tool (v4). KEGG pathway analysis and Gene Ontology (GO) analysis were performed using the clusterProfiler R package (v3.0.0) (Yu *et al*, 2012) and DAVID. To compare the level of similarity among the samples and their replicates, we used two methods: principal component analysis and Euclidean distance-based sample clustering. Gene set enrichment analysis was performed using GSEA (v3.0; Subramanian *et al*, 2005). All the downstream statistical analyses and generating plots were performed in R environment (v3.1.1; https://www.r-project.org/).

### Statistical analysis

Animal experiments were conducted unblinded. For certain readouts (e.g., determining *C. albicans* CFUs within organs), investigators were blinded. If not indicated otherwise, data are represented as mean ± SEM to provide an estimate of variation. No randomization was used. Results were analyzed with Prism version 8.0 (GraphPad software, Inc.). Statistical significance between groups was determined by using the two-tailed unpaired Student's *t*-test when a normal distribution was assumed or otherwise by Mann–Whitney *U*-test. Where indicated, data were analyzed by one-way ANOVA followed by Tukey's honestly significant difference (HSD) *post hoc* test for multiple comparisons. *P* values < 0.05 were considered significant; different levels of significance are indicated as follows: *$P$ < 0.05; **$P$ < 0.01; ***$P$ < 0.001.

# Data availability

The datasets produced in this study are available in the following databases. RNA-Seq data: Gene Expression Omnibus GSE149162 (https://www.ncbi.nlm.nih.gov/geo/query/acc.cgi?acc=GSE149162).

**Expanded View** for this article is available online.

## Acknowledgements

We thank the patients and their families for consenting to participate in this study; Gilad Evrony for help with genetic analyses; Thomas Scambler for help with T cell culture and calcium measurements and Michael Cammer for providing the ImageJ Script for analyzing ORAI1 and STIM1 co-localization. 29A2 anti-ORAI1 antibody was kindly provided by Dr. I. Kacskovics (Immuno-Genes). This study was funded by NIH grants AI097302 and AI137004 (S.F.) and AI065303 (D.U.), postdoctoral fellowships by the German Research Foundation (DFG) to S.K. (KA 4514/1-1), M.V. (VA 882/1-1), and U.K. (KA 4083/2-1), and a postdoctoral fellowship by the National Multiple Sclerosis Society to U.K. (FG-1608-25544). The Microscopy Laboratory at NYUSOM is partially supported by the Cancer Center Support Grant P30CA016087 of the Laura and Isaac Perlmutter Cancer Center; the Seahorse XFe24 analyzer was purchased with funding by NIH SIG grant 1S10OD016304-01.

## Author contributions

SK and SF designed the research and wrote the manuscript. SK, UK, ARC, LN, MV, LK, JY, JM, MM, PP, and PS conducted experiments. SK, UK, ARC, LN, MV, LK, JY, JM, MM, AK-J, DR, ZS, PP, PBS and PS, DU and SF analyzed data and interpreted the results. CF, SBC, and SET provided patient materials and data. SF supervised the study. All authors read and approved the final version of the manuscript.

## Conflict of interest

S.F. is a scientific cofounder of Calcimedica. The other authors declare that they have no conflict of interest.

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
