## [Review Process File · EMBO Molecular Medicine]

STIM1-mediated calcium influx controls antifungal immunity and the metabolic function of Th17 cells

Sascha Kahlfuss, Ulrike Kaufmann, Axel Concepcion, Lucile Noyer, Dimitrius Raphael, Martin Vaeth, Jun Yang, Mate Maus, James Muller, Lina Kozhaya, Alireza Khodadadi-Jamayran, Zhengxi Sun, Priya Pancholi, Patrick Shaw, Derya Unutmaz, Peter Stathopoulos, Cori Feist, Scott Cameron, Stuart Turvey, and Stefan Feske

DOI: 10.15252/emmm.201911592

Corresponding author(s): Stefan Feske (Stefan.Feske@nyulangone.org)

Review Timeline:

Submission Date:	11th Oct 19
Editorial Decision:	12th Nov 19
Revision Received:	20th Apr 20
Editorial Decision:	13th May 20
Revision Received:	19th May 20
Accepted:	25th May 20

Editor: Celine Carret

Transaction Report:

12th Nov 2019

Dear Dr. Feske,

Thank you for the submission of your manuscript to EMBO Molecular Medicine. We have now heard back from the three referees whom we asked to evaluate your manuscript.

As you will see in the comments pasted below, the referees find the study to be of interest and generally well performed and written. Unfortunately, several important concerns are raised that must be addressed as described. While ref. #1 mostly requests better explanations and clarifications, ref. #2 and #3 are more critical. Indeed, ref. #2 has reservations about the assumption that one STIM1 missense mutation can give the very complex and severe clinical picture. While possible, this referee would like to see more done to support that assumption, otherwise the conclusions should be tone down. This referee also insists on using *Candida* stimulation on Th17 cells as well as PMA+ionomycin. Ref. #3 has two main issues, the 1st one is about the model used. Since the Vav-Cre mouse model is available in the lab, cellular assays with bone-marrow derived cells should be used to document that neutrophils from the STIM1-deficient mice have decreased antifungal capacity. The 2nd point is about the mode of action of the STIM1 mutation on ORA1 activation at the plasma membrane.

Given these comments, we would therefore welcome the submission of a revised version within three to six months for further consideration and would like to encourage you to address all the criticisms raised as suggested to improve conclusiveness and clarity. Please note that EMBO Molecular Medicine strongly supports a single round of revision and that, as acceptance or rejection of the manuscript will depend on another round of review, your responses should be as complete as possible.

I look forward to receiving your revised manuscript.

Yours sincerely,

Celine Carret

Celine Carret, PhD | Senior Editor EMBO Molecular Medicine

Photos 400-800 DPI

*Additional important information regarding figures and illustrations can be found at <http://bit.ly/EMBOPressFigurePreparationGuideline>

***** Reviewer's comments *****

Referee #1 (Remarks for Author):

This is an important and elegant study that provides major new insight into the role of Th17 cells in orchestrating immune response to fungal infection. The authors find a novel mutation (L374P) in the second coiled coil domain of STIM1 that impairs store-operated calcium entry, T cell proliferation and cytokine production. They further demonstrate, using a combination of state-of-the-art techniques spanning the single molecule to in vivo rodent studies, a central role for STIM1 in regulating the metabolism of Th17 cells as well as antifungal immunity to systemic *C. albicans* infection. The study is carefully conducted, the data are convincing and the findings are novel and exciting.

I have only a few minor comments.

Fungi typically enter the body via mast cell-rich organs, such as the skin, gut, and airways. Mast cells respond to fungi both because of i) their strategic location at vascularized mucosal surfaces and ii) they express TLR2 and Dectin-1 receptors, activation of which release mediators known to be involved in antifungal responses. I wonder whether the authors looked for changes in mast cell numbers etc in the tongues of the infected mice? This is not essential for this study but the authors may already have the data.

The characterisation of the L374P STIM1 mutant is very rigorous. In Figure 2E, a clear smattering of mcherryL374P-STIM1 is seen in the TIRF images, which is not the case for the wild type mcherry STIM1. Some mcherryL374P-STIM1 clusters also appear to have formed under resting conditions. The authors explain this by drawing an analogy with their previous work on the R429C mutant, which destabilised CC3 structure and led to the exposure of the polybasic domain and subsequent accumulation in ER-PM junctions. However, looking at the data presented here and in Maus et al. I have the impression that there are more clusters/puncta of mcherryL374P-STIM1 at rest than was the case with the R429C mutant. I may of course be wrong but, if not, this might suggest that L347P promotes STIM1 clustering to some extent but prevents CAD from binding Orai1. Perhaps the authors could comment on this.

There is considerably more co-localization between mcherryL374P-SRTIM1 and Orai1-GFP than is the case for wild type STIM1 at rest. The distribution of Orai1 also looks a little strange to me; it seems to be in clumps and closely mirrors the distribution of mcherryL374P-STIM1. This is reflected in the Pearson coefficient graph in Figure 2E. Do the authors think this stronger co-localization under resting conditions is purely coincidental, reflecting the location of the mutant STIM1 at ER-

PM junctions or could there be some interaction with Orai1, albeit not strong enough to enable calcium entry?

In the methods, the authors state they measured the area under the curve to quantify calcium entry in P1, P2, mother and HD. This is shown in the left hand bar chart of Figure 1F. But what is analyzed in the right hand graph? The y-axis states Peak ratio but relative to what? The base line prior to TG stimulation or to the response in TG/0Ca just prior to readmission of external Ca. The way the data are presented gives the impression that there is no difference between P1 and P2 regarding SOCE, but the raw data show almost no SOCE at all for P2. It might be better to show the Peak ratio relative to the response in TG/0Ca just prior to readmission of external calcium.

Referee #2 (Remarks for Author):

The manuscript from Kahlfuss et al. describes two patients with a missense mutation in STIM1, and defects in Ca influx upon stimulation of T-cells, followed by fungal infections. Subsequently, the authors report increase in susceptibility to mucosal and systemic Candida infection in STIM1 knock-out mice.

Comments

1. The clinical picture of both patients is far more complex and severe than merely increase susceptibility to fungal infections. Indeed, extended morphological and neuro-muscular defects accompany the immunological phenotype. These defects are far more severe than it would be expected from an isolated defect in Th17 function, and most likely for a STIM1 missense mutation. The authors have focused on STIM1 sequencing, based on their earlier studies on the molecule, but the arguments that this is the only, and certainly the causative mutation, are missing. A number of missing pieces are necessary for a thorough genetic assessment of the family:
 - a. A comprehensive chromosomal and genetic analysis of the patients is needed and should be presented. Whole-exome/genome data would be needed, to assess the breadth of the genetic defects.
 - b. What is the frequency of this missense mutation in the general population? This information is crucial: a presence of the mutation in the general population of healthy individuals would invalidate the role of the mutation.
 - c. Ideally, a second family with this defect and the similar phenotype would be needed to validate this mutation as causative.
 - d. Mutation/disease segregation in the family is missing: the healthy sister and the father, who should not be homozygous of this mutation (as they are healthy). It is true that sometime it is difficult to recruit all members of a family, but that piece of information is very important to support the importance of this mutation.
2. In Figure 3 the stimulation of Th17 has been performed with PMA+ionomycin. This should be accompanied by direct stimulation with *Candida albicans*, which is an excellent inducer of Th17 cytokines: the release of IL-17 and IL-22 upon *Candida* stimulation should be shown.
3. How many mice were studied in Fig.4E, on the survival after disseminated candidiasis?
4. It is very difficult to understand the cause of death of mice with systemic *Candida* infection: they show very high mortality starting with day 8 post-infection (Fig.4E), but on day 6 only very few mice had any *Candida* in their organs (Fig.4H). Especially the majority of the control mice are almost free of fungal growth, with the exception of 1-2 mice. For mice who would succumb due to infection two days later, that is very strange. It is well known that in systemic candidiasis the target organ is the kidney in the mouse, and mice die of massive fungal infiltration and kidney insufficiency. How can

the authors explain this discrepancy?

5. The histology data in the kidney suggest hyperinflammation in the tissues, but this is a different pathophysiology than Th17 defects.

Referee #3 (Comments on Novelty/Model System for Author):

The mouse model does not adequately recapitulate the human disease

Referee #3 (Remarks for Author):

This study reports reduced antifungal CD4-mediated immunity in two patients with a point mutation in the ER Ca²⁺ sensor STIM1 and links this defect to a reduced metabolic function of non-pathogenic Th17 cells. A p.L347P mutation in the STIM1 channel activating domain was identified in two siblings with combined immunodeficiency suffering from recurrent bacterial and fungal infections. Store-operated Ca²⁺ entry was reduced despite normal STIM1 protein expression in the patients' T cells, which failed to expand and to secrete cytokines. When expressed in HEK cells, STIM1-L374P localizes to the TIRF plane prior to store depletion and fail to form clusters and to co-localize with ORAI1 upon store depletion. In mice, conditional STIM1 deletion in T cells increased the susceptibility to systemic, but not to mucosal infection with *Candida albicans*, increased the expression of non-pathogenic genes in Th17 cells and reduced their glycolytic and oxidative metabolic capacity. Glycolytic function and mitochondrial respiration was also impacted in CD4 T cells from a human patient. The authors conclude that STIM1 promotes antifungal immunity by regulating differentially the metabolism of pathogenic and non-pathogenic Th17 cells.

Comments: This is a well-controlled study that presents high quality data relevant for our understanding of the cellular basis of antifungal immunity. The data are solid, well presented, and for the most part adequately interpreted. The manuscript is also very well written. My enthusiasm is somewhat limited by the use of the CD4-cre mouse model and the exclusive focus on Th17 cells, which ignores the contribution of innate immune cells and thus fail to establish a solid link between the human disease and the mouse data. I also would suggest to better document the molecular defect imparted by the L347P mutation on STIM1 conformational changes during activation. Specifically:

1. It is difficult to relate the metabolic defect of Th17 cells to the increased susceptibility of the patients to fungal infections, because the mouse model does not adequately recapitulate the human disease. Mucosal infections cannot be reproduced in *Stim1^{fl/fl}-Cd4Cre* mice (Fig 4) yet can be readily generated in *Stim1^{fl/fl}-VavCre* mice (Fig S4). The choice of the Cd4-Cre and the subsequent focus on Th17 cells is questionable and ignores the major contribution of innate immune cells, whose role in antifungal defence is well established. In line with this, there is an increase in GM-CSF and neutrophils in the blood of infected *Stim1^{fl/fl}-Cd4Cre* mice. STIM1 was shown to regulate superoxide production by neutrophils (PMID:24493668, PMID:28724541), and one would thus expect that defective neutrophil functions contribute to the fungal infections observed in the two patients. The contribution of STIM1 in the antifungal response of neutrophils should be better documented.

2. The molecular defect induced by the STIM1-L347P mutation is not established. The authors state that STIM1-L347P fails to bind Orai1, but there is no evidence for this. Quite the contrary, Fig

2DE show that mCherry-STIM1-L347P has a high degree of co-localisation with overexpressed GFP-ORAI1 in the TIRF plane. This suggests that the two proteins co-localize at the ER-PM interface but fail to form clusters following store depletion. Lack of cluster formation cannot be used as evidence for defective STIM-ORAI binding, because cluster formation relies on the exposure of STIM1 polybasic tail rather than on binding to ORAI1. Several studies showed that STIM1 is first recruited to plasma membrane clusters via its polybasic tail and subsequently traps ORAI1 within clusters, resulting in channel activation (PMID: 25057023). This raises the possibility that STIM1-L347P is actually recruited to the PM via increased binding to ORAI1. Such a mutant would trap ORAI1 effectively, yet fail to cross-link and activate ORAI1 channels. Alternatively, STIM1-L347P might be pre-recruited to the PM via an exposed polybasic tail. To distinguish between these possibilities the authors should test whether STIM1-L347P is still recruited to the PM when expressed in the absence of ORAI1, and whether removing the polybasic tail impacts STIM1-L347P distribution. A co-IP is also required to document the binding of STIM1-L347P to endogenous and overexpressed ORAI1

Minor comments

In the text the mutation is referred to as STIM1 g.C1142T but in figure 1B and Supplementary Table 2 the mutation is T1121C. Based on p.L374P, the genetic mutation should be consistently g.T1142C.

Page 11, line 9 '...with the transcriptional identify of non-pathogenic TH-17 cells,..' should be 'identity'.

Point-by-Point response**Referee 1:**

This is an important and elegant study that provides major new insight into the role of Th17 cells in orchestrating immune response to fungal infection. The authors find a novel mutation (L374P) in the second coiled coil domain of STIM1 that impairs store-operated calcium entry, T cell proliferation and cytokine production. They further demonstrate, using a combination of state-of-the-art techniques spanning the single molecule to in vivo rodent studies, a central role for STIM1 in regulating the metabolism of Th17 cells as well as antifungal immunity to systemic C. albicans infection. The study is carefully conducted, the data are convincing and the findings are novel and exciting. I have only a few minor comments.

Response: We thank the reviewer for his/her thoughtful and positive evaluation of our study and comments that have improved the manuscript.

Fungi typically enter the body via mast cell-rich organs, such as the skin, gut, and airways. Mast cells respond to fungi both because of i) their strategic location at vascularized mucosal surfaces and ii) they express TLR2 and Dectin-1 receptors, activation of which release mediators known to be involved in antifungal responses. I wonder whether the authors looked for changes in mast cell numbers etc in the tongues of the infected mice? This is not essential for this study but the authors may already have the data.

Response: We agree that mast cells are increasingly recognized to play a role in antifungal immunity (Jiao, Luo et al., 2019, Renga, Moretti et al., 2018). It is therefore tempting to speculate that overall mast cell numbers and/or function is altered in our STIM-deficient mice during systemic and/or local candida infection. In addition, it was recently demonstrated that mast cell function in house dust mite-mediated asthmatic airway inflammation depends on store-operated calcium entry (SOCE) (Lin, Nelson et al., 2018). It is therefore possible that SOCE in mast cells could be involved in antifungal immunity at mucosal surfaces. Although STIM1 and STIM2 are deleted in mast cells of *Stim1^{fl/fl}VaviCre* and *Stim1^{fl/fl}Stim2^{fl/fl}VaviCre* mice and we agree that it would be interesting to study the role of SOCE in mast cells in antifungal immunity, we had to prioritize experiments for the revision of this manuscript and focused on the role of SOCE neutrophils (as requested by reviewer 3). A detailed analysis of SOCE in mast cells will be subject of a future study.

The characterisation of the L374P STIM1 mutant is very rigorous. In Figure 2E, a clear smattering of mcherryL374P-STIM1 is seen in the TIRF images, which is not the case for the wild type mcherry STIM1. Some mcherryL374P-STIM1 clusters also appear to have formed under resting conditions. The authors explain this by drawing an analogy with their previous work on the R429C mutant, which destabilised CC3 structure and led to the exposure of the polybasic domain and subsequent accumulation in ER-PM junctions. However, looking at the data presented here and in Maus et al. I have the impression that there are more clusters/puncta of mcherryL374P-STIM1 at rest than was the case with the R429C mutant. I may of course be wrong but, if not, this might suggest that L347P promotes STIM1 clustering to some extent but prevents CAD from binding Orai1. Perhaps the authors could comment on this.

Response: Based on our analysis of the change in localization of STIM1 to the TIRF evanescent field (Figure 2G) and the colocalization of ORAI1 with STIM1 (Figure 2H) derived from the TIRF images in Figure 2D,E, we conclude that the STIM1 p.L374P mutation and the previously reported STIM1 p.R429C mutation behave very similarly (Maus, Jairaman et al., 2015). To illustrate this point, we here provide a side-by-side comparison of TIRF experiments in which HEK cells were transfected with these mutants of STIM1.

There is considerably more co-localization between mcherryL374P-SRTIM1 and Orai1-GFP than is the case for wild type STIM1 at rest. The distribution of Orai1 also looks a little strange to me; it seems to be in clumps and closely mirrors the distribution of mcherryL374P-STIM1. This is reflected in the Pearson coefficient graph in Figure 2E. Do the authors think this stronger co-localization under resting conditions is purely coincidental, reflecting the location of the mutant STIM1 at ER-PM junctions or could there be some interaction with Orai1, albeit not strong enough to enable calcium entry?

Response: The reviewer brings up an interesting point (whether the "stronger co-localization under resting conditions reflects the location of the mutant STIM1 at ER-PM junctions or if there could be some interaction with Orai1") that was also raised by Reviewer 3. We have therefore conducted additional experiments to address the following questions: (1) is the localization of mutant STIM1 to the plasma membrane mediated by the effects of the L374P mutation on the release of the polybasic domain (PBD) of STIM1 or is it dependent on residual binding to ORAI1? To this end, we used TIRFM to analyze cells expressing a STIM1 mutant lacking the PBD and cells lacking ORAI1. (2) Does the L374P mutation impair STIM1 binding to ORAI1? To this end, we conducted co-IP experiments. A detailed description of the results addressing questions 1 and 2 is provided in our response to Reviewer 3 and **new Supplemental Figure 2**.

In the methods, the authors state they measured the area under the curve to quantify calcium entry in P1, P2, mother and HD. This is shown in the left hand bar chart of Figure 1F. But what is analyzed in the right hand graph? The y-axis states Peak ratio but relative to what? The base line prior to TG stimulation or to the response in TG/0Ca just prior to readmission of external Ca. The way the data are presented gives the impression that there is no difference between P1 and P2 regarding SOCE, but the raw data show almost no SOCE at all for P2. It might be better to show the Peak ratio relative to the response in TG/0Ca just prior to readmission of external calcium.

Response: As suggested by Reviewer 1, we have reanalyzed the calcium data and now show the peak ratio relative to the response in TG/0 calcium prior to readmission of external calcium in Figure 1F.

Referee 2:

The manuscript from Kahlfuss et al. describes two patients with a missense mutation in STIM1, and defects in Ca influx upon stimulation of T-cells, followed by fungal infections. Subsequently, the authors report increase in susceptibility to mucosal and systemic Candida infection in STIM1 knock-out mice.

Comments

1. The clinical picture of both patients is far more complex and severe than merely increase susceptibility to fungal infections. Indeed, extended morphological and neuro-muscular defects accompany the immunological phenotype. These defects are far more severe than it would be expected from an isolated defect in Th17 function, and most likely for a STIM1 missense mutation. The authors have focused on STIM1 sequencing, based on their earlier studies on the molecule, but the arguments that this is the only, and certainly the causative mutation, are missing.

Response: We thank the reviewer for his/her careful evaluation of our manuscript and comments. We agree that P1 and P2 suffer from a complex disease not only involving the immune system or Th17 cells. We do not claim that the *STIM1* mutation is only affecting Th17 cell function but it clearly has additional effects in other tissues. We and others have described mutations in *STIM1* (and in the CRAC channel encoding gene *ORAI1*), which result in a complex disease that we named CRAC channelopathy (OMIM 612782 and 612783). Hallmarks of CRAC channelopathy are (i) combined immunodeficiency syndrome with recurrent bacterial, viral and fungal infections, (ii) autoimmunity with autoantibodies against platelets and RBCs, (iii) ectodermal dysplasia with anhidrosis and amelogenesis imperfecta type III, and (iv) muscular hypotonia (Lacruz & Feske, 2015). The *STIM1* p.L374P mutant patients reported in this manuscript have the same or very similar phenotype as other patients with LOF mutations in *STIM1* (or *ORAI1*). The causal relationship between the *STIM1* p.L374P mutation and the observed clinical phenotype is therefore not just based on the 2 patients reported here, but also on the previous reports of *STIM1* loss-of-function (LOF) mutations and the resulting clinical CRAC channelopathy phenotype. We have provided a table summarizing these studies (**Reviewer Table 1**) to illustrate the common disease phenotype resulting from *STIM1* (and *ORAI1*) mutations.

To address the concerns of Reviewer 2, we have performed additional genetic analyses, which are summarized below and have been added to the manuscript text.

A number of missing pieces are necessary for a thorough genetic assessment of the family:

a. A comprehensive chromosomal and genetic analysis of the patients is needed and should be presented. Whole-exome/genome data would be needed, to assess the breadth of the genetic defects.

Response: We agree that in general a comprehensive chromosomal and genetic analysis is required when a mutation in a gene not previously reported in connection with a clinical phenotype is suggested to be causative for that disease. In the case of the patients reported here, we do not think that a WES/WGS analysis is necessary because of the arguments presented above, namely that the clinical phenotype of P1 and P2 (who are homozygous for the *STIM1* p.L374P mutation) is practically identical to that of other patients with LOF mutations in *STIM1* (**Reviewer Table 1**). We think that this provides sufficient evidence that the *STIM1* p.L374P mutation is responsible for the disease in the patients reported here.

b. What is the frequency of this missense mutation in the general population? This information is crucial: a presence of the mutation in the general population of healthy individuals would invalidate the role of the mutation.

Response: We thank the reviewer for his/her comment as we had not adequately addressed this in the previous version of the manuscript. We performed additional genetic analyses: (1) gnomAD Analysis (<https://gnomad.broadinstitute.org/>) using the gnomAD v3 data set (GRCh38 / hg38) from unrelated individuals sequenced as part of various disease-specific and population genetic studies. The input term (gene name) “*STIM1*” and a search for the variants “p.Leu374Pro” and “c.1121T>C” yielded that the

p.Leu374Pro *STIM1* mutation is extremely rare with an allele frequency of 6.98e-6 (1 allele out of 143330).

Gene	Mutation	Immuno-deficiency	Auto-immunity	Muscular hypotonia	Ectodermal dysplasia		References
					Anhidrosis	Amelogenesis imperfecta	
ORAI1	p.R91W	Y	n.r.	Y	n.r.	n.r.*	Feske et al. (1996); Feske et al. (2006)
	p.R91W	Y	Y	Y	Y	Y	Feske et al. (2000); Feske et al. (2006)
	p.V181SfsX8	Y	n.r.	Y	Y	Y	Lian et al. (2018)
	p.L194P	Y	Y	Y	Y	*	Lian et al. (2018)
	p.G98R	Y	Y	Y	Y	Y	Lian et al. (2018)
	p.G98R	Y	Y	Y	Y	Y	Lian et al. (2018)
	p.A88SfsX25	Y	Y	Y	Y	n.r.*	Partiseti et al. (1994); McCarl et al. (2009)
	p.A103E/p.L194P	Y	n.r.	Y	Y	Y	Le Deist et al. (1995); McCarl et al. (2009)
	p.H165PfsX1	Y	n.r.	Y	n.r.	n.r.*	Chou et al. (2015)
	p.R270X	Y	n.r.	Y	n.r.	n.r.*	Badran et al. (2016)
STIM1	p.E128RfsX9	Y	Y	Y	n.r.	Y	Picard et al. (2009)
	p.E128RfsX9	Y	Y	Y	n.r.	Y	Picard et al. (2009)
	C1538-1 G>A	Y	n.r.	Y	n.r.	n.r.	Byun et al. (2010)
	p.R429C	Y	Y	Y	Y	Y	Fuchs et al. (2012), Maus et al. (2015)
	p.R429C	Y	Y	Y	Y	Y	Fuchs et al. (2012), Maus et al. (2015)
	p.R426C	n.r.	n.r.	n.r.	Y	Y	Wang et al. (2014)
	p.P165Q	Y	Y	Y	Y	Y	Schaballie et al. (2015)
	p.P165Q	Y	n.r.	Y	Y	Y	Schaballie et al. (2015)
	p.L74P	Y	Y	n.r.	Y	Y	Parry et al. (2016)
	p.L74P	Y	n.r.	n.r.	Y	Y	Parry et al. (2016)
	p.L374P (P1)	Y	n.r.	Y	Y	Y	This study
p.L374P (P2)	Y	Y	Y	Y	Y	This study	

Reviewer Table 1: CRAC channelopathy due to mutations in the genes *ORAI1* and *STIM1*. *STIM1* mutations at the bottom of the table are those reported in this manuscript. Abbreviations: n.r., not reported or not tested; N, No; Y, Yes. * Patients died before complete dentition.

As "CRAC channelopathy" due to mutations in the genes *ORAI1* or *STIM1* follows an autosomal recessive inheritance, the homozygous p.Leu374Pro mutation in *STIM1* detected in both patients in the current study

is compatible with their disease ('CRAC channelopathy'). (2) We also calculated the scaled Combined Annotation-Dependent Depletion (CADD) score for the *STIM1* c.1121T>C, p.Leu374Pro mutation by using the Single Nucleotide Variant (SNV) lookup tool <https://cadd.gs.washington.edu/snv>. The (maximum) CADD score at this position is 28.5. (Request: Chromosome 11, Position 4103565, CADD GRCh37-v1.4), which indicates that the mutation is within the top 0.1% of the most deleterious variants in the human genome. We now report these findings in the manuscript.

c. Ideally, a second family with this defect and the similar phenotype would be needed to validate this mutation as causative.

Response: We agree that studying a second family with the same mutation and phenotype would be ideal. However, mutations in *STIM1* (and *ORAI1*) are extremely rare and no other family with the exact same *STIM1* p.L374P mutation is known. However there are several other patients and families with LOF mutations in *STIM1* that present with the same clinical phenotype (**Reviewer Table 1**), which in our opinion validates the *STIM1* p.L374P mutation as causative of the disease in our patients. (We would like

to point out that the novelty of our study is not the disease phenotype itself but the in-depth analysis of the role STIM1 and SOCE in antifungal immunity, which has never been reported before.)

d. Mutation/disease segregation in the family is missing: the healthy sister and the father, who should not be homozygous of this mutation (as they are healthy). It is true that sometime it is difficult to recruit all members of a family, but that piece of information is very important to support the importance of this mutation.

Response: We agree that the father and sister II.3 should not be homozygous for the mutation. They are clinical healthy (as far as we could ascertain) and do not show any hallmark symptoms of CRAC channelopathy. Unfortunately, neither the sister nor the father (who is estranged from the rest of the family and incarcerated) are available for genetic testing.

2. In Figure 3 the stimulation of Th17 has been performed with PMA+ionomycin. This should be accompanied by direct stimulation with Candida albicans, which is an excellent inducer of Th17 cytokines: the release of IL-17 and IL-22 upon Candida stimulation should be shown.

Response: We thank the reviewer for this comment and have conducted the suggested experiments. First, we have cultured expanded human T cells from patient 1, his mother and two healthy donors (HD) in the presence of *C. albicans* for 2 and 4 weeks in vitro and restimulated T cells with PMA and ionomycin (P+I) to induce production of Th17 cytokines. In addition, we treated HD T cells with the selective CRAC channel inhibitor GSK-7975 (Rice, Bax et al., 2013, Wen, Voronina et al., 2015) for 6 hours during the restimulation of T cells with P+I. We found that CD4+ and CD8+ T cells from patient 1 and T cells from HDs treated with GSK-7975 have an almost complete defect in the production of IL-17A (and IL-2) compared to T cells from the mother and a HD. We were not able to detect IL-22 production in either HD or patient cells. Second, we cultured PBMC from a HD for 6 days in the presence of *C. albicans* and then restimulated cells for 6 hours with P+I in the presence of GSK-7975 or DMSO as control. CD4+ and CD8+ T cells showed a near complete defect in the production of IL-17A. We now show the experiments proposed by the reviewer within a new **Figure 4** as we think they strengthen our findings. Third, it is noteworthy that T cells from P1 and P2 that were stimulated with *C. albicans* and analyzed for T cell proliferation as part of their clinical evaluation for immunodeficiency showed a severe defect in *C. albicans*-specific proliferation (**Supplemental Table 2**).

3. How many mice were studied in Fig.4E, on the survival after disseminated candidiasis?

Response: Five *Stim1^{fl/fl}Cd4Cre* and 12 WT mice were used for these experiments. We have added this information to the manuscript. We thank the reviewer for drawing our attention on this omission.

4. It is very difficult to understand the cause of death of mice with systemic Candida infection: they show very high mortality starting with day 8 post-infection (Fig.4E), but on day 6 only very few mice had any Candida in their organs (Fig.4H). Especially the majority of the control mice are almost free of fungal growth, with the exception of 1-2 mice. For mice who would succumb due to infection two days later, that is very strange. It is well known that in systemic candidiasis the target organ is the kidney in the mouse, and mice die of massive fungal infiltration and kidney insufficiency. How can the authors explain this discrepancy?

Response: *Stim1^{fl/fl}Cd4Cre* mice started to succumb to systemic *C. albicans* infection at day 8 p.i. and at day 17 p.i. all mice had died (**revised Figure 6E**). The fungal burden (CFU per gram of infected organs) at day 6 p.i. show high CFU counts in 3 *Stim1^{fl/fl}Cd4Cre* mice, which correspond to the mice that succumbed to *C. albicans* early post-infection (**revised Figure 6H**). *Stim1^{fl/fl}Cd4Cre* mice with lower fungal burdens at day 6 p.i. died at later time points. The same correlation between fungal burdens and time of death was observed for WT mice. At day 6 p.i. only 2 WT mice showed high fungal burdens and

these mice died at day 10 or 11 p.i. The remaining WT mice that had no detectable *C. albicans* in their kidneys at day 6 p.i. (**Figure 6H**) died between days 11 and 20 or survived the systemic *C. albicans* infection. To further illustrate the increased fungal burdens in the kidneys of *Stim1^{fl/fl}Cd4Cre* mice compared to WT littermate controls, we added new Periodic acid–Schiff (PAS) stains to detect polysaccharides in the cell wall of *C. albicans* in the kidneys of *Stim1^{fl/fl}Cd4Cre* and WT mice at day 6 post-infection (**new Figure 6I**, right panels). Taken together, we think that the increased fungal burdens of *Stim1^{fl/fl}Cd4Cre* mice compared to WT controls correlates well with their faster and more complete (100%) death from systemic *C. albicans* infection.

While answering reviewer #2's comment, we did however notice a mistake in the x-axis labeling of the original **Figure 4F** (now **revised Figure 6F**), which we corrected. Fungal burdens were measured at day 6p.i.

5. The histology data in the kidney suggest hyperinflammation in the tissues, but this is a different pathophysiology than Th17 defects.

Response: This is an interesting point and we welcome the opportunity to comment on it. The kidney histologies of *C. albicans* infected *Stim1^{fl/fl}Cd4Cre* mice indeed show increased leukocyte infiltration (inflammation) compared to WT mice (revised **Figure 6I**). This finding correlates well with the higher numbers of Cd11b⁺ Gr-1⁺ neutrophils in the blood of infected *Stim1^{fl/fl}Cd4Cre* mice (revised **Figure 6K**) and the increased frequencies of GM-CSF-producing CD4⁺ T cells in the spleen of these mice (revised **Figure 6J**) compared to infected WT littermates. These results are also consistent with our RNA sequencing data of *in vitro* differentiated non-pathogenic Th17 cells. *Stim1^{fl/fl}Cd4Cre* non-pathogenic Th17 cells show significantly elevated expression of *Csf2* compared to WT Th17 cells (revised **Figure 7C**). We think that this enhanced GM-CSF production by CD4⁺ T cells of *Stim1^{fl/fl}Cd4Cre* mice is responsible for neutrophil mobilization from the bone marrow, which are subsequently recruited to the *C. albicans* infected kidneys of these mice. However, the lack of many other important Th17 (and Th1) effector cytokines in *Stim1^{fl/fl}Cd4Cre* mice (**Figure 6J**) may result in an overall impaired innate and adaptive immune response to systemic *C. albicans* infection compared to WT mice.

A similar kidney inflammation was recently described in *Il17ra^{ΔRTEC}* mice with conditional deletion of the IL-17 receptor A (IL-17RA) in renal tubular epithelial cells (RTEC). 7 days after systemic infection with *C. albicans* the authors of this study noted "increased tissue pathology in the cortex and inner and outer medullary regions" consistent on H&E staining of serial kidney sections with leukocyte infiltration; see Figure 5A of (Ramani, Jawale et al., 2018). Systemic *C. albicans* infection in these mice resulted in impaired kidney function and death of mice by 10 days p.i. similar to findings in *Stim1^{fl/fl}Cd4Cre* mice after systemic *C. albicans* infection. We have added a sentence to the discussion of our manuscript mentioning the kidney inflammation in *Stim1^{fl/fl}Cd4Cre* and *Il17ra^{ΔRTEC}* mice after systemic *C. albicans* infection.

Referee 3:

*This study reports reduced antifungal CD4-mediated immunity in two patients with a point mutation in the ER Ca²⁺ sensor STIM1 and links this defect to a reduced metabolic function of non-pathogenic Th17 cells. A p.L347P mutation in the STIM1 channel activating domain was identified in two siblings with combined immunodeficiency suffering from recurrent bacterial and fungal infections. Store-operated Ca²⁺ entry was reduced despite normal STIM1 protein expression in the patients' T cells, which failed to expand and to secrete cytokines. When expressed in HEK cells, STIM1-L374P localizes to the TIRF plane prior to store depletion and fail to form clusters and to co-localize with ORAI1 upon store depletion. In mice, conditional STIM1 deletion in T cells increased the susceptibility to systemic, but not to mucosal infection with *Candida albicans*, increased the expression of non-pathogenic genes in Th17 cells and reduced their glycolytic and oxidative metabolic capacity. Glycolytic function and mitochondrial respiration was also impacted in CD4 T cells from a human patient. The authors conclude that STIM1 promotes antifungal immunity by regulating differentially the metabolism of pathogenic and non-pathogenic Th17 cells.*

Comments: This is a well-controlled study that presents high quality data relevant for our understanding of the cellular basis of antifungal immunity. The data are solid, well presented, and for the most part adequately interpreted. The manuscript is also very well written.

Response: We thank the reviewer for his/her careful evaluation of our manuscript and positive assessment.

My enthusiasm is somewhat limited by the use of the CD4-cre mouse model and the exclusive focus on Th17 cells, which ignores the contribution of innate immune cells and thus fail to establish a solid link between the human disease and the mouse data. I also would suggest to better document the molecular defect imparted by the L347P mutation on STIM1 conformational changes during activation.

Response: To address the reviewers two main points we performed additional experiments and provide additional discussion within the manuscript. **(1)** Regarding the contribution of innate immune cells to antifungal immunity, we completely agree with the reviewer and have included additional data and restructured the manuscript (see below). **(2)** Regarding the molecular defect caused by the STIM1 p.L374P mutation, we have conducted additional experiments, which were added **new Supplemental Figure 2** and are discussed in detail below.

Specifically: 1. It is difficult to relate the metabolic defect of Th17 cells to the increased susceptibility of the patients to fungal infections, because the mouse model does not adequately recapitulate the human disease. Mucosal infections cannot be reproduced in Stim1fl/fl-Cd4Cre mice (Fig 4) yet can be readily generated in Stim1fl/fl-VavCre mice (Fig S4). The choice of the Cd4-Cre and the subsequent focus on Th17 cells is questionable and ignores the major contribution of innate immune cells, whose role in antifungal defence is well established. In line with this, there is an increase in GM-CSF and neutrophils in the blood of infected Stim1fl/fl-Cd4Cre mice. STIM1 was shown to regulate superoxide production by neutrophils (PMID:24493668, PMID:28724541), and one would thus expect that defective neutrophil functions contribute to the fungal infections observed in the two patients. The contribution of STIM1 in the antifungal response of neutrophils should be better documented.

Response: We completely agree with the reviewer regarding the important role of innate immune cells for antifungal immunity. We therefore decided to include our data showing impaired immunity to mucosal infection with *C. albicans* in *Stim1^{fl/fl}* and *Stim1/2^{fl/fl}-Vav-iCre* mice to the main figures (**new Figure 5**). Since *Stim1^{fl/fl}* and *Stim1/2^{fl/fl}-Vav-iCre* mice lack SOCE in all immune cells, they better recapitulate the immunodeficiency in human patients, who also lack SOCE in all immune cells. Our data show that *Stim1^{fl/fl}* and *Stim1/2^{fl/fl}-Vav-iCre* mice show strongly increased susceptibility to mucosal infection with *C. albicans*, which is in line with the mucocutaneous *Candida* infections observed in several patients with CRAC channelopathy (**Table 1**).

We also agree with the reviewer's comment that "*STIM1 was shown to regulate superoxide production by neutrophils [...] and one would thus expect that defective neutrophil functions contributes to the fungal infections observed in the two patients*". We have therefore conducted new experiments to test the role of SOCE in ROS production and candidicidal function of neutrophils. Using poly-I:C injected *Stim1^{fl/fl}* and *Stim1/2^{fl/fl}-Mx1-Cre* mice, which lack SOCE in all immune cells (similar to *Stim1^{fl/fl}* and *Stim1/2^{fl/fl}-Vav-iCre* mice), we found impaired ROS production in the absence of STIM1 or both STIM1/STIM2 and decreased fungal killing after coincubation of *C. albicans* with STIM1 or STIM1/STIM2 deficient neutrophils. We added these results to **new Figure 5H,I** and have updated the results and discussion sections of the manuscript accordingly.

We would like to briefly comment on two additional points raised by the reviewer: (1) "*Mucosal infections cannot be reproduced in Stim1fl/fl-Cd4Cre mice (Fig 4) yet can be readily generated in Stim1fl/fl-VavCre mice (Fig S4)*." The finding that *Stim1^{fl/fl}Cd4Cre* mice are not more susceptible to mucosal *Candida* infection than WT mice (revised **Figure 6A-D**) is consistent with studies showing that mice lacking T cells are not significantly more susceptible to mucosal *Candida* infection than WT mice either (Conti, Peterson et al., 2014, Gladiator, Wangler et al., 2013). (2) "*It is difficult to relate the metabolic defect of Th17 cells to the increased susceptibility of the patients to fungal infections.*" We here show that Th17 cell function of

Stim1^{fl/fl}Cd4Cre mice is impaired, which makes these mice more susceptible to systemic *Candida* infection. This is in line with the recently reported role of Th17 cells in mediating immunity to systemic *Candida* infection (Shao, Ang et al., 2019). Our data show that STIM1 is important for Th17 cell function in antifungal immunity, and that STIM1 is critical for regulating Th17 cell metabolism as metabolic pathways were among the most deregulated pathways in STIM1-deficient Th17 cells as determined by RNA-Seq analysis.

2. The molecular defect induced by the STIM1-L347P mutation is not established. The authors state that STIM1-L347P fails to bind Orai1, but there is no evidence for this. Quite the contrary, Fig 2DE show that mCherry-STIM1-L347P has a high degree of co-localisation with overexpressed GFP-ORAI1 in the TIRF plane. This suggests that the two proteins co-localize at the ER-PM interface but fail to form clusters following store depletion. Lack of cluster formation cannot be used as evidence for defective STIM-ORAI binding, because cluster formation relies on the exposure of STIM1 polybasic tail rather than on binding to ORAI1. Several studies showed that STIM1 is first recruited to plasma membrane clusters via its polybasic tail and subsequently traps ORAI1 within clusters, resulting in channel activation (PMID: 25057023). This raises the possibility that STIM1-L347P is actually recruited to the PM via increased binding to ORAI1. Such a mutant would trap ORAI1 effectively, yet fail to cross-link and activate ORAI1 channels. Alternatively, STIM1-L347P might be pre-recruited to the PM via an exposed polybasic tail. To distinguish between these possibilities the authors should test whether STIM1-L347P is still recruited to the PM when expressed in the absence of ORAI1, and whether removing the polybasic tail impacts STIM1-L347P distribution. A co-IP is also required to document the binding of STIM1-L347P to endogenous and overexpressed ORAI1

Response: We agree with the reviewer that it is interesting to study how the L374P mutation abolishes SOCE despite being localized at the PM and partially colocalizing with ORAI1. We had initially not planned a more comprehensive analysis to dissect the effects of the L374P mutation because (i) the paper is mostly focused on the role of SOCE in immunity to *C. albicans* and (ii) because the effects of the STIM1 p.L374P mutation are similar to those of a STIM1 p.R429C mutation we had reported earlier (Maus et al., 2015). To respond to the reviewer's request we have, however, conducted several additional experiments to address the concerns raised. The results are summarized in **new Supplemental Figure 2**.

(1) We tested the hypothesis that mutant STIM1 is localized at the PM because the L374P mutation results in an "exposed polybasic tail" which traps STIM1 at ER-PM junctions. To this end we used HEK293 cells stably expressing GFP-ORAI1 and transfected them with mCherry-tagged wildtype STIM1, STIM1 lacking the polybasic domain (Δ K), STIM1 p.L374P or STIM1 p.L374P- Δ K. We analyzed the localization the behaviour of these STIM1 proteins with regard to localization at the PM, cluster (puncta) formation and colocalization with ORAI1 by total internal reflection fluorescence microscopy (TIRFM). As shown in **new Supplemental Figure 2A-C**, deletion of the polybasic domain does not affect the TG-induced translocation of STIM1- Δ K to the PM, puncta formation or colocalization with ORAI1 (**panel B**), which is consistent with (Park, Hoover et al., 2009), who also showed that STIM1- Δ K forms puncta in cells overexpressing ORAI1 as in our experiments. As shown in our original data (Figure 2E), we again found here that STIM1 p.L374P is prelocalized at the PM and partially colocalizes with ORAI1 (**panel B**). Deletion of the polybasic domain had no significant effect on the behaviour of STIM1 p.L374P- Δ K, which was still localized at the PM and partially colocalized with ORAI1, suggesting that the localization of the STIM1 mutant is not dependent on its polybasic domain.

(2) We next tested the hypothesis that mutant STIM1 is localized at the PM because of binding to ORAI1. To this end we transfected HEK293 cells in which ORAI1 was deleted by CRISPR/Cas9 gene editing with either WT STIM1 or the three mutant STIM1 constructs described above and analyzed the localization of STIM1 at the PM by TIRFM. As shown in the **Figure added for the reviewers below**, deletion of ORAI1 abolished the translocation of WT STIM1 to the PM after thapsigargin stimulation. Importantly, deletion of ORAI1 does not prevent the prelocalization of STIM1 p.L374P at the PM. Even combined deletion of ORAI1 and the polybasic domain of STIM1 p.L374P- Δ K had no significant effect on

the localization of mutant STIM1 at the PM. This finding is unexpected but indicates that the localization of mutant STIM1 p.L374P at ER-PM junctions is independent of its polybasic domain and ORAI1. At present, we can only speculate about the mechanism. It is possible that STIM1 p.L374P has a partially active configuration which allows it to oligomerize with endogenous STIM1 or STIM2 in HEK293 cells and piggybacks on them to translocate to ER-PM junctions. This is plausible, as coexpression of STIM2 with STIM1 Δ K in HEK293 cells was shown to result in coclustering of both proteins suggesting that STIM2 recruits STIM1 to ER-PM junctions (Ong, de Souza et al., 2015). We planned to test this idea by deleting STIM2 in HEK293 cells and overexpressing STIM1 p.L374P- Δ K, but these experiments were stopped when our lab was shut down due to the COVID19 outbreak.

(3) Finally, we tested whether mutant STIM1 is able to bind to ORAI1 to respond to the reviewer's comments "*a co-IP is also required to document the binding of STIM1-L347P to endogenous and*

[Figure for reviewers removed upon authors request]

overexpressed ORAI1". To this end, we cotransfected HEK293 cells with Flag-ORAI1 and either WT or mutant STIM1 p.L374P. Cells were left unstimulated or stimulated with thapsigargin in the absence of extracellular Ca²⁺ to maximally deplete ER stores and induce STIM1 activation. ORAI1 was immunoprecipitated with anti-Flag beads and STIM1 binding detected by SDS-PAGE using an anti-STIM1 antibody. Our results from three repeat experiments show that even in unstimulated cells WT STIM1 can be co-IP'ed with ORAI1, and as expected this co-IP increased when cells were stimulated with thapsigargin (**new Supplemental Figure 2D,E**). The STIM1 p.L374P mutant showed stronger co-IP with ORAI1 in unstimulated cells. This increased co-IP is consistent with the constitutive localization of STIM1 p.L374P at the PM we show by TIRFM in Figure 2 and S2. Since this constitutive PM localization of STIM1 p.L374P does not depend on ORAI1 (as shown in the Figure for reviewers) we speculate that the constitutive co-IP of STIM1 p.L374P and ORAI1 may be facilitated by the recruitment of STIM1 p.L374P to the PM by endogenous STIM1 or STIM2 as discussed above. In other words, the recruitment of mutant STIM1 to the PM does not depend on ORAI1, but once it is at the PM it is able to bind to ORAI1. After stimulation with thapsigargin to deplete ER stores, the co-IP between STIM1 p.L374P and ORAI1 was strongly reduced to levels similar to those found for WT STIM1 in unstimulated cells (**new Supplemental Figure 2D,E**). This result was unexpected because TG stimulation has no effect on the PM localization of STIM1 p.L374P (as shown in the Figure for reviewers). However, according to our model that STIM1 p.L374P may be recruited to the PM by endogenous STIM1 or STIM2, we speculate that TG-induced store depletion results in increased binding of endogenous STIM1 and STIM2 to ORAI1, which compete with and replace mutant STIM1 p.L374P. As stated above, we planned to test this idea by deleting STIM2 in HEK293 cells and repeating the co-IP experiments, but this was no longer possible when our lab was shut down due to the COVID19 outbreak.

Taken together, we propose the following **model** how the p.L374P mutant affects STIM1 function: L374P results in a conformational change in the C-terminus of STIM1 that partially activates it and allows STIM1 to translocate to the PM. This recruitment is not dependent on the polybasic domain of STIM1 or the presence of ORAI1 in the PM, at least not under conditions of ectopic overexpression of STIM1 p.L374P. We speculate that overexpressed STIM1 p.L374P binds to endogenous STIM1 or STIM2, which bring it to the PM and facilitate its binding to ORAI1. This binding, observed by partial colocalization in TIRFM and co-IP experiments, is weak and not sufficient to activate CRAC channel opening as apparent from (a) impaired SOCE and T cell function, and (b) reduced co-IP of ORAI1-STIM1 p.L374P after thapsigargin stimulation, likely because mutant STIM1 is outcompeted by ORAI1 binding of endogenous STIM1 or STIM2. What we cannot fully explain is why the mutation retains its ability to bind to ORAI1 but fails to activate it. One explanation is that CRAC channel activation requires the formation of STIM1 clusters (Luik, Wang et al., 2008). Whereas STIM1 p.L374P is located near the PM and binds ORAI1, it fails to form puncta after store depletion with thapsigargin, suggesting that the mutation interferes with proper oligomerization of STIM1 molecules, which is required for the formation of macromolecular complexes that have the correct STIM1:ORAI1 stoichiometry needed to activate the CRAC channel (Yen & Lewis, 2019). An additional explanation for impaired SOCE could be that the L374P mutation results in conformational changes that do not abrogate STIM1 binding to ORAI1 but specifically interfere with its ability to gate the CRAC channel. This interpretation is supported by the finding that the binding of mutant STIM1 to ORAI1 observed in co-IP experiments is lost after cell stimulation, likely because it is replaced by active endogenous WT STIM1 that has a higher affinity for ORAI1. This changed or weaker binding implies that the interaction of mutant STIM1 to ORAI1 may be different in a fundamental way and prevents proper ORAI1 gating.

Minor comments

In the text the mutation is referred to as STIM1 g.C1142T but in figure 1B and Supplementary Table 2 the mutation is T1121C. Based on p.L374P, the genetic mutation should be consistently g.T1142C.

Response: We thank the reviewer for pointing out this mistake which we have corrected in the text. The correct annotation of the mutation in P1 and P2 is c.1121T>C and p.Leu374Pro.

Page 11, line 9 '...with the transcriptional identity of non-pathogenic TH-17 cells,..' should be 'identity'.

Response: We have corrected this mistake.

Note that the RNA-Seq data shown in Figure 7 and 8 were reanalyzed to account for an updated version of R (v. 3.6.2) and the KEGG database (2019; the previous analysis was done using the 2016 version of KEGG accessed through DAVID). The reanalysis confirmed the results we had presented in the original manuscript, but resulted in slightly more accurate lists of differentially expressed genes (DEG) and KEGG pathways.

REFERENCES

Chen X, Kozhaya L, Tastan C, Placek L, Dogan M, Horne M, Abblett R, Karhan E, Vaeth M, Feske S, Unutmaz D (2018) Functional Interrogation of Primary Human T Cells via CRISPR Genetic Editing. *J Immunol* 201: 1586-1598

Conti HR, Peterson AC, Brane L, Huppler AR, Hernandez-Santos N, Whibley N, Garg AV, Simpson-Abelson MR, Gibson GA, Mamo AJ, Osborne LC, Bishu S, Ghilardi N, Siebenlist U, Watkins SC, Artis D, McGeachy MJ, Gaffen SL (2014) Oral-resident natural Th17 cells and gammadelta T cells control opportunistic *Candida albicans* infections. *J Exp Med* 211: 2075-84

Gladiator A, Wangler N, Trautwein-Weidner K, LeibundGut-Landmann S (2013) Cutting edge: IL-17-secreting innate lymphoid cells are essential for host defense against fungal infection. *J Immunol* 190: 521-5

Jiao Q, Luo Y, Scheffel J, Zhao Z, Maurer M (2019) The complex role of mast cells in fungal infections. *Exp Dermatol* 28: 749-755

Lacruz RS, Feske S (2015) Diseases caused by mutations in ORAI1 and STIM1. *Ann N Y Acad Sci* 1356: 45-79

Lin YP, Nelson C, Kramer H, Parekh AB (2018) The Allergen Der p3 from House Dust Mite Stimulates Store-Operated Ca(2+) Channels and Mast Cell Migration through PAR4 Receptors. *Mol Cell* 70: 228-241 e5

Luik RM, Wang B, Prakriya M, Wu MM, Lewis RS (2008) Oligomerization of STIM1 couples ER calcium depletion to CRAC channel activation. *Nature* 454: 538-42

Maus M, Jairaman A, Stathopoulos PB, Muik M, Fahrner M, Weidinger C, Benson M, Fuchs S, Ehl S, Romanin C, Ikura M, Prakriya M, Feske S (2015) Missense mutation in immunodeficient patients shows the multifunctional roles of coiled-coil domain 3 (CC3) in STIM1 activation. *Proc Natl Acad Sci U S A* 112: 6206-11

Ong HL, de Souza LB, Zheng C, Cheng KT, Liu X, Goldsmith CM, Feske S, Ambudkar IS (2015) STIM2 enhances receptor-stimulated Ca(2+)(+) signaling by promoting recruitment of STIM1 to the endoplasmic reticulum-plasma membrane junctions. *Sci Signal* 8: ra3

Park CY, Hoover PJ, Mullins FM, Bachhawat P, Covington ED, Raunser S, Walz T, Garcia KC, Dolmetsch RE, Lewis RS (2009) STIM1 clusters and activates CRAC channels via direct binding of a cytosolic domain to Orai1. *Cell* 136: 876-90

Ramani K, Jawale CV, Verma AH, Coleman BM, Kolls JK, Biswas PS (2018) Unexpected kidney-restricted role for IL-17 receptor signaling in defense against systemic *Candida albicans* infection. *JCI Insight* 3

Renga G, Moretti S, Oikonomou V, Borghi M, Zelante T, Paolicelli G, Costantini C, De Zuani M, Vilella VR, Raia V, Del Sordo R, Bartoli A, Baldoni M, Renauld JC, Sidoni A, Garaci E, Maiuri L, Pucillo C, Romani L (2018) IL-9 and Mast Cells Are Key Players of *Candida albicans* Commensalism and Pathogenesis in the Gut. *Cell Rep* 23: 1767-1778

Rice LV, Bax HJ, Russell LJ, Barrett VJ, Walton SE, Deakin AM, Thomson SA, Lucas F, Solari R, House D, Begg M (2013) Characterization of selective Calcium-Release Activated Calcium channel blockers in mast cells and T-cells from human, rat, mouse and guinea-pig preparations. *Eur J Pharmacol* 704: 49-57

Shao TY, Ang WXG, Jiang TT, Huang FS, Andersen H, Kinder JM, Pham G, Burg AR, Ruff B, Gonzalez T, Khurana Hershey GK, Haslam DB, Way SS (2019) Commensal *Candida albicans* Positively Calibrates Systemic Th17 Immunological Responses. *Cell Host Microbe* 25: 404-417 e6

Wen L, Voronina S, Javed MA, Awais M, Szatmary P, Latawiec D, Chvanov M, Collier D, Huang W, Barrett J, Begg M, Stauderman K, Roos J, Grigoryev S, Ramos S, Rogers E, Whitten J, Velicelebi G, Dunn M, Tepikin AV et al. (2015) Inhibitors of ORAI1 Prevent Cytosolic Calcium-Associated Injury of Human Pancreatic Acinar Cells and Acute Pancreatitis in 3 Mouse Models. *Gastroenterology* 149: 481-92 e7

Yen M, Lewis RS (2019) Numbers count: How STIM and Orai stoichiometry affect store-operated calcium entry. *Cell Calcium* 79: 35-43

13th May 2020

Dear Prof. Feske,

Thank you for the submission of your revised manuscript to EMBO Molecular Medicine. We have now received the enclosed reports from the referees that were asked to re-assess it. As you will see the reviewers are now supportive and I am pleased to inform you that we will be able to accept your manuscript pending the following final editorial amendments:

1) Please provide a point-by-point letter including my comments and your detailed responses to their comments (as Word file).

2) Please carefully check the authors guidelines for formatting your supplemental information:

Expanded view and Appendix (see:

<https://www.embopress.org/page/journal/17574684/authorguide#expandedview>)

Please label your supplemental information document "Appendix" and change the nomenclature as "Appendix figure S1", "Appendix Table S1" and so on. Remove the graphical abstract from this file. Include a Table of content on the 1st page. Have all text black. This document must be provided as pdf, it won't be edited nor typeset.

3) Figures: In fig. 6I H&E, if the lower and higher magnification are taken from the same tissues, please provide the origin boxes.

4) Source Data:

We encourage the publication of source data, particularly for electrophoretic gels, blots, but also microscopy images with the aim of making primary data more accessible and transparent to the reader. Would you be willing to provide a PDF file per figure that contains the original, uncropped and unprocessed scans of all or key gels used in the figure (including molecular weight markers)? The PDF files should be labeled with the appropriate figure/panel number (1 file/figure), and should have molecular weight markers; further annotation may be useful but is not essential. The PDF files will be published online with the article as supplementary "Source Data" files. If you have any questions regarding this just contact me.

5) In the main manuscript file, please do the following:

- correct/answer the track changes suggested by our data editors by working from the attached/uploaded document
- add up to 5 keywords
- remove the blue coloured text
- in M&M, the statistical paragraph should reflect all information that you have filled in the Authors checklist, especially regarding randomisation, blinding, replication.
- indicate in legends exact $n=$ and exact $p=$ values, not a range, along with the statistical test used. Some people found that to keep the figures clear, providing an Appendix table Sx with all exact p -values was preferable. You are welcome to do this if you want to.
- in M&M, for animal work, provide gender and age for all the animals used in each experiment.
- remove "data not shown" pp. 7,8,12. As per our guidelines, on "Unpublished Data" the journal does not permit citation of "Data not shown". All data referred to in the paper should be displayed in the main or Expanded View figures. "Unpublished observations" may be referred to in exceptional

cases, where these are data peripheral to the major message of the paper and are intended to form part of a future or separate study, the names of the persons that reported the observation should be listed in brackets. Personal communications (Author name(s), personal communications) must be authorised in writing by those involved, and the authorisation sent to the editorial office at time of submission.

- remove the word "disclosure" from "Conflict of interest disclosure"
- add Table 1 at the end of the manuscript not in a separate document.

6) Authors' contribution: the contributions of Dimitrius Raphael, Mate Maus, and Zhengxi Sun are missing

7) 4) For more information: There is space at the end of each article to list relevant web links for further consultation by our readers. Could you identify some relevant ones and provide such information as well? Some examples are patient associations, relevant databases, OMIM/proteins/genes links, author's websites, etc...

8) The Paper Explained: EMBO Molecular Medicine articles are accompanied by a summary of the articles to emphasize the major findings in the paper and their medical implications for the non-specialist reader. Please provide a draft summary of your article highlighting

- the medical issue you are addressing, = Problem
- the results obtained = Results
- their clinical impact = Impact

9) Every published paper now includes a 'Synopsis' to further enhance discoverability. Synopses are displayed on the journal webpage and are freely accessible to all readers. They include a short stand first (maximum of 300 characters, including space) as well as 2-5 one sentence bullet points that summarise the paper. Please write the bullet points to summarise the key NEW findings. They should be designed to be complementary to the abstract - i.e. not repeat the same text. We encourage inclusion of key acronyms and quantitative information (maximum of 30 words / bullet point). Please use the passive voice. Please attach these in a separate file or send them by email, we will incorporate them accordingly.

10) As part of the EMBO Publications transparent editorial process initiative (see our Editorial at <http://embomolmed.embopress.org/content/2/9/329>), EMBO Molecular Medicine will publish online a Review Process File (RPF) to accompany accepted manuscripts.

In the event of acceptance, this file will be published in conjunction with your paper and will include the anonymous referee reports, your point-by-point response and all pertinent correspondence relating to the manuscript. Let us know whether you agree with the publication of the RPF and as here, if you want to remove or not any figures from it prior to publication.

11) Data availability: To list the primary data generated in your study, we would kindly ask you to include a formal "Data availability" section (after Materials & Methods) that follows the example below:

- [data type]: [full name of the resource] [accession number/identifier] ([doi or URL or identifiers.org/DATABASE:ACCESSION])

example

* RNA-Seq data: Gene Expression Omnibus GSExxxxx

(<https://www.ncbi.nlm.nih.gov/geo/query/acc.cgi?acc=GSExxxxx>)

Please add the accession number now and make sure that the data is available as soon as the paper is accepted.

Please submit your revised manuscript within two weeks. I look forward to seeing a revised form of your manuscript as soon as possible.

Yours sincerely,

Celine Carret

Celine Carret, PhD
Senior Editor
EMBO Molecular Medicine

*** Instructions to submit your revised manuscript ***

To submit your manuscript, please follow this link:

Link Not Available

1) a .doc formatted version of the manuscript text (including Figure legends and tables)

2) Separate figure files*

3) supplemental information as Expanded View and/or Appendix. Please carefully check the authors guidelines for formatting Expanded view and Appendix figures and tables at <https://www.embopress.org/page/journal/17574684/authorguide#expandedview>

4) a letter INCLUDING the reviewer's reports and your detailed responses to their comments (as Word file).

***** Reviewer's comments *****

Referee #2 (Remarks for Author):

[The authors have responded adequately and the article is suitable for publication].

Referee #3 (Remarks for Author):

The authors have addressed all the points raised in the first round of review. The new data clarify the role of innate immune cells in the pathogenesis of the fungal infection. They also provide new insights into the mechanisms of activation of the mutant protein. The authors have to be congratulated for their fine work in addressing the comments. I have no further suggestion for change.

Point-by-Point response

Editor: *Thank you for the submission of your revised manuscript to EMBO Molecular Medicine. We have now received the enclosed reports from the referees that were asked to re-assess it. As you will see the reviewers are now supportive and I am pleased to inform you that we will be able to accept your manuscript pending the following final editorial amendments:*

Referee #2 (Remarks for Author):

[The authors have responded adequately and the article is suitable for publication].

Referee #3 (Remarks for Author):

The authors have addressed all the points raised in the first round of review. The new data clarify the role of innate immune cells in the pathogenesis of the fungal infection. They also provide new insights into the mechanisms of activation of the mutant protein. The authors have to be congratulated for their fine work in addressing the comments. I have no further suggestion for change.

Response: *We thank all 3 reviewers for their time and expertise used to review this manuscript, which has helped to improve the quality of the published work.*

25th May 2020

Dear Prof. Feske,

Thank you for revising your article as suggested. We are pleased to inform you that your manuscript is accepted for publication and is now being sent to our publisher to be included in the next available issue of EMBO Molecular Medicine.

Please read below for additional IMPORTANT information regarding your article, its publication and the production process.

Congratulations on your interesting work,

Celine Carret

Celine Carret, PhD
Senior Editor
EMBO Molecular Medicine

Follow us on Twitter @EmboMolMed
Sign up for eTOCs at embopress.org/alertsfeeds

Corresponding Author Name: Dr. Stefan Feske
 Journal Submitted to: EMBO Molecular Medicine
 Manuscript Number: EMM-2019-11592